# The Carcinogen Cadmium Activates Lysine 63 (K63)-Linked Ubiquitin-Dependent Signaling and Inhibits Selective Autophagy

**DOI:** 10.3390/cancers13102490

**Published:** 2021-05-20

**Authors:** Abderrahman Chargui, Amine Belaid, Papa Diogop Ndiaye, Véronique Imbert, Michel Samson, Jean-Marie Guigonis, Michel Tauc, Jean-François Peyron, Philippe Poujeol, Patrick Brest, Paul Hofman, Baharia Mograbi

**Affiliations:** 1Université Côte d’Azur, Institute of Research on Cancer and Aging in Nice (IRCAN), Centre National de la Recherche Scientifique (CNRS), Institut National de la Santé et de la Recherche Médicale (INSERM), Fédération Hospitalo-Universitaire (FHU) OncoAge, Centre Antoine Lacassagne, F-06189 Nice, France; cherguiman@yahoo.fr (A.C.); abelaid@bu.edu (A.B.); papadiogop.ndiaye@gmail.com (P.D.N.); patrick.brest@univ-cotedazur.fr (P.B.); hofman.p@chu-nice.fr (P.H.); 2Higher School of Agriculture of Kef, University Jendouba, Le Kef and Laboratory of Histology, Embryology and Cell Biology, Faculty of Medicine Tunis, 7110 Le Kef, Tunisia; 3Université Côte d’Azur, Centre Méditerranéen de Médecine Moléculaire (C3M), Institut National de la Santé et de la Recherche Médicale (INSERM), F-06204 Nice, France; Veronique.IMBERT@univ-cotedazur.fr (V.I.); Jean-Francois.PEYRON@univ-cotedazur.fr (J.-F.P.); 4Université Côte d’Azur, Laboratory Transporter in Imaging and Radiotherapy in Oncology (TIRO), Direction de la Recherche Fondamentale (DRF), Institut des sciences du vivant Fréderic Joliot, Commissariat à l’Energie Atomique et aux énergies alternatives (CEA), F-06107 Nice, France; michelsamson64@yahoo.com (M.S.); Jean-Marie.GUIGONIS@univ-cotedazur.fr (J.-M.G.); 5Université Côte d’Azur, Laboratoire de Physiomédecine Moléculaire, LP2M, Labex ICST, Centre National de la Recherche Scientifique (CNRS), F-06107 Nice, France; Michel.TAUC@univ-cotedazur.fr (M.T.); philippe.poujeol@yahoo.com (P.P.); 6Université Côte d’Azur, Laboratory of Clinical and Experimental Pathology, FHU OncoAge, Hospital-Integrated Biobank (BB-0033-00025), Centre Hospitalier Universitaire (CHU) de Nice, F-06001 Nice, France

**Keywords:** K63-linked ubiquitination, tumor suppression, selective autophagy, starvation-induced autophagy, aggrephagy, CYLD deubiquitinase, carcinogen, cadmium, bafilomycin A1, NF-κB

## Abstract

**Simple Summary:**

Environmental exposure to cadmium (Cd) is associated with cancer. Cadmium was classified in 1993 by the International Agency for Research on Cancer as a carcinogen. However, as no biological process requires Cd, the molecular mechanisms of its carcinogenicity remain an enigma. This study is the first, to our knowledge, to show that Cd exposure has an impact on K63 ubiquitination, selective autophagy, and signaling pathways of lung and kidney epithelial cells, three crucial drivers of cancer initiation and progression. Further clarification of these issues will provide valuable insight into prognostic biomarkers and/or therapeutic targets for cancer.

**Abstract:**

Signaling, proliferation, and inflammation are dependent on K63-linked ubiquitination—conjugation of a chain of ubiquitin molecules linked via lysine 63. However, very little information is currently available about how K63-linked ubiquitination is subverted in cancer. The present study provides, for the first time, evidence that cadmium (Cd), a widespread environmental carcinogen, is a potent activator of K63-linked ubiquitination, independently of oxidative damage, activation of ubiquitin ligase, or proteasome impairment. We show that Cd induces the formation of protein aggregates that sequester and inactivate cylindromatosis (CYLD) and selective autophagy, two tumor suppressors that deubiquitinate and degrade K63-ubiquitinated proteins, respectively. The aggregates are constituted of substrates of selective autophagy—SQSTM1, K63-ubiquitinated proteins, and mitochondria. These protein aggregates also cluster double-membrane remnants, which suggests an impairment in autophagosome maturation. However, failure to eliminate these selective cargos is not due to alterations in the general autophagy process, as degradation of long-lived proteins occurs normally. We propose that the simultaneous disruption of CYLD and selective autophagy by Cd feeds a vicious cycle that further amplifies K63-linked ubiquitination and downstream activation of the NF-κB pathway, processes that support cancer progression. These novel findings link together impairment of selective autophagy, K63-linked ubiquitination, and carcinogenesis.

## 1. Introduction

The conjugation of a single or multiple ubiquitin (Ub) molecules to a protein, known as ubiquitination, regulates cell proliferation, differentiation, transformation, and death [1,2]. This broad range of functions most probably stems from the myriad of polyubiquitin chains formed through any of seven lysines of ubiquitin (K6, K11, K27, K29, K33, K48, and K63) [3]. The different linkages have unique functions: the most described is a chain of four or more Ub, which are linked through lys48 (K48-polyUb) that targets proteins for proteasome degradation, and thereby to signal termination [4,5]. In contrast, the role of K63-polyUb that serves signaling functions is only beginning to be explored. As with phosphorylation, K63-linked ubiquitination can activate protein kinases, dictate cellular localization, and serve as a docking site for protein-protein interactions during DNA repair [6], activation of the nuclear factor κB (NF-κB) pathway [7], and selective autophagy [8,9].

A key issue is how the abundance of the K63-linked Ub chain is limited to very low levels. High K63-linked ubiquitination is known to lead to neurodegeneration, inflammation, and cancer [1]. K63-deubiquitinases (A20, OTULIN, and CYLD…) and macroautophagy (hereafter referred to as autophagy) degrade K63-linked ubiquitinated proteins and organelles [8,9,10]. As autophagy progresses, cytosolic constituents are surrounded by the phagophore. The phagophore then expands to form a double-membrane vesicle, termed an autophagosome. The completed autophagosome fuses with a lysosome in which the cargo is degraded. As knowledge into autophagy has developed, an increasing number of autophagy substrates have been identified. Macroautophagy was thought for decades to be a bulk, non-selective self-eating process that degrades long-lived proteins to cope with starvation and other environmental challenges. In addition to this fundamental role, basal autophagy is involved in the turnover of unneeded or damaged proteins/organelles that accumulate [11,12]. The triage decision is dictated by K63-linked ubiquitination, which helps recruit the autophagy adaptor SQSTM1/p62 that binds simultaneously to microtubule-associated protein 1 light chain 3 (LC3) on the nascent autophagosome, thereby ensuring selective sequestration and degradation of the targeted substrates [13]. Thus, the picture now emerging is that constitutive autophagy selectively degrades K63-linked ubiquitinated substrates and is often called selective autophagy [8,9]. However, whether or not starvation-induced autophagy can clear ubiquitinated proteins is still questionable.

The hypothesis that autophagy suppresses tumor development was first put forward by the finding that *BECN1* is deleted in 40% to 75% of breast, ovarian, colon, and prostate cancers. Consistently, allelic loss of *Becn1* was demonstrated to predispose mice to lymphomas, hepatocellular carcinomas, and lung carcinomas [14,15]. Likewise, defects in other autophagy genes (*Atg4c*, *Atg5, Atg7*, *Uvrag*, *Ambra1*, and *Sh3glb1*) render cells or mice prone to developing tumors [12,16]. As a tumor suppression mechanism, autophagy may maintain genome stability, induce senescence, and limit inflammation in normal cells and early-stage carcinomas [16,17]. At the molecular level, we and others have provided evidence that autophagy can switch off several key oncogenic signaling pathways (mTOR, Wnt, NF-κB, RHOA) by degrading kinases (SRC, RET, KIT), G proteins (RHOA), scaffold proteins (SQSTM1, inflammasome), as well as transcription factors (RELA/p65, BCL10, IKBKB, SNAIL1, TWIST, etc.). We therefore propose the term “SIGNALphagy” to indicate a dedicated type of macroautophagy by which a cell can control its own growth, survival, motility, and inflammatory response [18,19].

However, as a tumor develops, autophagy drives cancer progression by supplying metabolic substrates, buffering oxidative damage, promoting chemoresistance, tumor metastasis, and maintenance of the cancer stem cell pool [16,20]. To reconcile these opposing findings, it has been proposed that the differential roles of autophagy in cancer may be related to tissue and stage specificities. However, we assume instead that these two types of autophagy, selective or bulk autophagy, may degrade different substrates and thereby serve different functions in cancer. For possible future therapeutic application, we need to understand which type of autophagy, selective or bulk autophagy, ensures tumor suppression/promotion. All the genetic or pharmacological strategies used so far inhibit both autophagy pathways. In addition, all types of autophagy rely on the same autophagy machinery and the same pathway. By inference, they are assumed to degrade the same long-lived proteins, but this hypothesis remains elusive.

Therefore, to address this issue, we explored the possibility that a carcinogen might specifically disrupt the type of autophagy involved in tumor suppression. We chose to study cadmium (CdCl_2_, Cd), a human carcinogen and a major health hazard for 10% of the world population [21,22,23,24]. Inhalation of polluted air, or cigarette smoke, and ingestion of contaminated water/diet have been linked to human diseases. Poorly excreted Cd has a long biological half-life (20 years) and cannot be degraded, and thus accumulates in the body. The kidney is the most sensitive organ that contains 60% of the Cd body burden as the proximal convoluted tubule (PCT) accounts for the majority of the uptake of Cd [25,26]. The other major Cd target organs are the lungs, which explains the shortness of breath and respiratory failure in severe cases [21,22,23,24].

The tumorigenicity of chronic low doses of Cd is associated with lung cancer, the leading cause of cancer-related deaths worldwide, and to a lesser degree to renal carcinoma [27]. Molecularly, the underlying mechanism involves the intracellular accumulation of Cd and then its ability to displace other essential trace metals from critical proteins, mostly unknown. Despite the so-called ionic mimicry concept [28], the carcinogenic power of Cd remains elusive, as Cd is a weak mutagen at chronic low environmental doses. However, Cd-treated cells have consistently shown oxidative stress, disruption of cell signaling pathways, accumulations of ubiquitinated proteins, and autophagic vesicles [29,30,31]. This prompted us to use renal epithelial and lung cancer cells to explore the effect of Cd on the autophagy pathway.

Our study uncovers novel and critical consequences of Cd exposure at low subtoxic doses: Cd is the first example of a carcinogen that induces K63-linked ubiquitination. We propose that K63 ubiquitination resulted in part from the ability of Cd to mimic Zn and alert selective autophagy, a Zn sensing pathway that degrades K63-Ub proteins [5,9,32,33,34,35]. One of the major events following Cd exposure is the accumulation of SQSTM1, a selective K63-Ub autophagy receptor, and an NF-κB signaling scaffold, which contains a Zn binding (ZZ) domain [13,36,37,38]. Downstream, K63-Ub/SQSTM1 accumulation results in the sequestration of damaged mitochondria, the CYLD K63 DUB, and the activation of the K63-Ub TRAF6-NF-κB pathway, providing metabolic, survival, inflammatory, and proliferative advantages to tumor cells (Figure 1). Together, these findings present a new paradigm linking disruption of selective autophagy, signaling through K63-linked polyubiquitination, and carcinogenesis.

## 2. Materials and Methods

### 2.1. Cell Culture and Treatments

Mouse PCT (proximal convoluted tubules), monkey kidney epithelial COS7, and human lung A549 cancer cell lines (ATCC) were used. Their physiological origins correspond to the common routes of human Cd exposure (oral, inhalation).

As there is no suitable human in vitro PCT model that replicates the high degree of sensitivity to Cd (HK-2, or HEK293 cells, 40 μM [39,40,41,42]), we established a murine immortalized non-cancerous PCT cell line [43]. Our experimental model fully recapitulates the changes observed in vivo in normal kidneys following exposure to very low effective Cd environmental doses (5 μM [29]). 

We chose the following subtoxic Cd doses that produced maximum ubiquitination: PCT (Cd 5 μM), COS7 (Cd 10 µM), and A549 (Cd 10–20 µM). For all experiments, cells were serum-starved for 30 min in fresh serum-free DMEM/F12 supplemented with ITS-A (Invitrogen, Thermo Fisher Scientific, Paris, France) and treated with Cd (CdCl_2_; Sigma-Aldrich, Saint-Quentin Fallavier, France), alone or in combination with reactive oxygen species (ROS) scavengers (NAC (N-acetyl cysteine; 10 mM), vitamin E (100 µM)), proteasome (MG132; 10 µM, Sigma), or IKK-2 (sc-514; 100µM, Calbiochem, MERCK, Guyancourt, France) inhibitors. As a control, cells were untreated or stimulated with LPS (Lipopolysaccharide; 5 µg/mL, Sigma), FCS (fetal calf serum; 2%), HgCl_2_ (5 µM), Pb (NO_3_)2 (5 µM), CoCl_2_ (100 µM), FeCl_2_ (100 µM), or Zn(NO_3_)_2_ (100 µM). To inhibit the last step in autophagic degradation, cells were treated with a specific inhibitor of v-ATPase activity, bafilomycin A_1_ (bafA1; 100 nM, Sigma), or a weak base that raises the intralysosomal pH, chloroquine (CQ; 100 µM, Sigma).

### 2.2. Detection of Oxidative Stress

The intracellular content of glutathione (GSH) was quantified using the glutathione assay kit (Sigma) as described [43]. Intracellular reactive oxygen species (ROS) levels were also measured in parallel using carboxy-H_2_DCFDA (10 μM; carboxy-2’,7’-dichlorodihydrofluorescein diacetate; Molecular Probes), a probe that produces a fluorescent signal when oxidized by ROS, most notably hydrogen peroxide. Briefly, PCT cell lines were loaded for 30 min with these probes and incubated in the presence of either Cd (5 μM), NAC (10 mM), or vitamin E (100 µM), alone or in combination. Hydrogen peroxide (H_2_0_2_, 1 μM) was used as a positive control. Variations in the cell fluorescence were measured every 2 min using a Genius spectrofluorimeter (SAFAS, Monaco, Monaco). Accumulation of oxidized (i.e., carbonylated) proteins was detected by reaction with 2,4-dinitrophenylhydrazine followed by immunoblotting with an anti-DNP antibody (OxyBlot^™^ kit, Sigma-Aldrich, MERCK). No signal was obtained when all reagents except DNP-hydrazine were added to cell lysates.

### 2.3. Cell Lysis

Cells (ø 100, 5 × 10^6^ cells, 70% confluency) were treated with Cd, washed in PBS, and solubilized in ice-cold lysis buffer (50 mM Tris-HCl, pH 7.5, 0.5% Triton-X100, phosphatase inhibitors (1 mM Na_3_VO_4_, 10 mM NaF, 10 mM βglycerophosphate, 25 mM NaPPi), 1 mM EDTA, and protease inhibitors (Complete ^TM^; Sigma-Aldrich)) for 30 min at 4 °C. Soluble and insoluble fractions were recovered following centrifugation (15,000× *g*, 15 min, 4 °C), and the insoluble pellets were then resuspended in Laemmli buffer by sonication. All buffers (lysis, Laemmli, and wash) were supplemented in fresh 1 mM *N*-ethyl-maleimide (NEM, DUB inhibitor).

### 2.4. Immunoblotting

Whole-cell lysates (WCL), immunoprecipitates, soluble, and insoluble fractions (40–75 µg) were analyzed by immunoblotting, with antibodies that examine the expression of autophagy markers and substrates (the lipid-conjugated LC3-II form (clone 5F10; Nanotools), and SQSTM1 (human, #610833; BD Transduction Laboratories™, Le Pont de Claix, France or mouse, SC25575, Santa Cruz Biotechnology, Clinisciences, Nanterre, France), as previously described [17]. Downstream, for the detection of ubiquitinated proteins, immunoblots were incubated with antibodies that specifically recognize total ubiquitinated (DAKO; 1/6000), pan polyubiquitinated (Chemicon; FK2, 1/7500), and K63-linked polyubiquitinated (BIOMOL; HWA4C4, 1/7500) proteins. Alternatively, the key NF-κB proteins and targets (TRAF6 (Santa Cruz Biotechnology), the Ser^32^/Ser^36^-phosphorylated IκBa (P-IκBa, Cell Signaling Technology), and CCND1/Cyclin D1 (Cell Signaling Technology)) were also examined by immunoblotting. As we were unable to detect CYLD in Cd-treated PCT cells with available antibodies, we analyzed the effect of Cd on the subcellular distribution of CYLD by transfection of COS7 cells with CYLD-FLAG (Gilles Courtois, INSERM U1038, CEA, Grenoble, France). After overnight incubation at 4 °C and three washes with TNT (10 min, 10 mM Tris HCl, pH 7.4, 0.15 M NaCl, 0.1% Tween 20), the bound primary antibodies were revealed with a horseradish peroxidase-conjugated anti-mouse antibody or anti-rabbit (Santa Cruz Biotechnology, 1:10,000, 1 h) and visualized with the Immobilon Classico Western HRP substrate (MERCK). Equal loadings of proteins were verified by staining the blot with amido black or by re-probing the same blots with anti-ACTIN or anti-TUBULIN (Santa Cruz Biotechnology).

### 2.5. Peptide Preparation for Mass Spectrometry(MS) Analysis

MS was used to identify the type of ubiquitin linkage. PCT cells (from 20 ø100 mm culture dishes) were stimulated with Cd (5 µM, 6 h), washed twice, and collected by scraping into ice-cold PBS. The cells were centrifuged at 800× *g* for 5 min, resuspended in 5 mL of homogenization buffer (3 mM imidazole, pH 7.4, and 250 mM sucrose supplemented with phosphatase inhibitors), and disrupted with 30 strokes of a tight-fitting pestle of a Dounce homogenizer. The post-nuclear supernatant fraction was prepared by centrifugation of the lysates at 1000× *g* for 10 min. After centrifugation at 24,000× *g* for 10 min, the pellet containing insoluble ubiquitinated proteins was resuspended in 50 mM ammonium bicarbonate by sonication and digested overnight at 37 °C with trypsin (sequencing grade modified trypsin; Promega; ref V5111) at a ratio protein/enzyme of 20. To prepare the internal standard Ub peptides, K48-linked and K63-linked tetraubiquitins (K48Ub_4_ and K63Ub_4_; Boston Biochem, R&D systems, Lille, France) were subjected to the same protocol of digestion with trypsin.

### 2.6. Mass Spectrometry

Analyses of the tryptic peptides were carried out using a cap-LC system (Ultimate 3000 RSLC system, Dionex) coupled to a mass spectrometer Q-Exactive Plus Orbitrap (Thermofisher Scientific). Briefly, the peptides were separated onto a C18 column (WATERS, ACQUITY UPLC M-Class C18, 130 Å pore size, 300 × 150 μm, 1.7 μm) by a 5–45% linear gradient of solvent B (80% acetonitrile, 0.1% formic acid) against solvent A (0.1% formic acid) over 180 min at a flow rate of 2 μL/min. Scan mass spectra were acquired from 350 to 1500 m/z with the automatic gain control (AGC) target set to 3 × 10^6^ and a resolution of 70,000 (Orbitrap). A top 15 data-dependent method was used for MS/MS spectrum acquisition with an AGC target of 1 × 10^5^, a resolution of 35 K, and a dynamic exclusion of 40 s.

All resulting MS/MS spectra were analyzed with Proteome Discoverer software (Xcalibur version 2.0.7, Thermo Scientific, Waltham, MA, USA), using the Sequest search engine against the Uniprot Mouse database (Version 2015_2). Precursor mass tolerance was set to 10 ppm, and fragment ion tolerance was 0.02 Da. The modification of ubiquitination on Lysine (+383.228 Da) was set in dynamic mode. Verification of ubiquitination was also evaluated using Mascot Server 2.3 software (Matrix Science). A decoy database search strategy was also used to estimate the false discovery rate (FDR) to ensure the reliability of the proteins identified, and at least two peptides were required for matching a protein entry for its identifications.

### 2.7. Proteasome Activity

To examine the chymotrypsin-like activity of the proteasome, PCT cells were treated with Cd (6 h), lysed using proteasome lysis buffer (10 mM Tris-HCl, pH 7.5, 1% Triton-X100, 150 mM NaCl, 0.5 mM EDTA, 1 mM DTT, 5 mM MgCl_2_, 0.5 mM PMSF, 1 µg/mL Pepstatin, 1 µg/mL Leupeptin, 1 µg/mL Aprotinin), followed by centrifugation at 12,000× *g* for 10 min. The proteasomal activity in cell extracts (25 µg) was assessed by measuring the hydrolysis of a fluorogenic substrate Suc-LLVY-Amc (80 µM; λexcitation: 380 nm, λemission: 460 nm; Sigma) in 200 μL of assay buffer (10 mM Tris-HCl, pH 7.5, 0.5 mM EDTA, 1 mM DTT, 5 mM MgCl_2_, 2 mM ATP) for 1 h at 37 °C. The specificity of the reaction was confirmed by preincubating cell lysates with an excess of the proteasomal inhibitor, MG132 (10 µM). Results are the means of duplicates from three experiments.

### 2.8. In Vitro Ubiquitination and Deubiquitination Assays

PCT cells (4 ø 100 mm) were sonicated in 1 mL of ice-cold PBS. After centrifugation (20,000× *g,* 10 min), the supernatant (containing E1, E2, E3 ubiquitin ligases, and deubiquitinating enzymes) was divided into aliquots and stored at −80 °C.

In vitro ubiquitination assay: Cell extracts (30 µg) were incubated with 5 µg of ubiquitin (Boston Biochem), 2 mM MgCl_2_, 5 mM ATP, 5 mM phosphocreatine, and 1 U of creatine kinase in reaction buffer (20 mM HEPES, pH 7.5; 2 mM DTT). Since lysates also contain DUB enzymes and proteasomes that degrade Ub proteins generated during the reactions, lysates were preincubated with ubiquitin aldehyde (20 μg/mL; DUB inhibitor; Boston Biochem) and MG132 (10 µM; proteasome inhibitor) to facilitate detection of ubiquitinated proteins. The reactions were performed in the absence of ATP (–ATP), where indicated. After incubation at 37 °C for 16 h in the presence of Cd (5 µM), the reaction was terminated by the addition of Laemmli buffer, boiled at 95 °C for 5 min, and resolved by 15% SDS–PAGE, and analyzed by FK2 or K63 polyUb immunoblotting.

In vitro de-ubiquitination assay: Cell extracts (30 µg) were preincubated for 20 min at room temperature with or without Cd (5 µM) or NEM (DUB inhibitor, 10 mM), and 0.25 µg of K48Ub_4_ or K63Ub_4_ were then added and incubated in 20 µL DUB assay buffer (20 mM Tris-HCl, pH 7.5, 5 mM MgCl_2_, 2 mM DTT) at 30 °C for 16 h. The reaction was terminated by the addition of Laemmli buffer, resolved by 15% SDS-PAGE, and analyzed by immunoblotting with anti-ubiquitin.

### 2.9. Detection of Autophagy

Four hallmarks monitored the effect of Cd on the autophagy pathway: (i) the formation of autophagic vesicles (mRFP-EGFP-LC3 and transmission electron microscopy) and the degradation of four well-established autophagy substrates and cargos, (ii) membrane-associated LC3-II, (iii) SQSTM1, (iv) mitochondria, and (v) long-lived proteins.

### 2.10. Electron Microscopy

The ultrastructural changes caused by Cd in PCT cells (2 and 6 h) were analyzed by transmission electron microscopy, as previously described [44]. Twenty to twenty-five micrographs, primary magnification ×10,000, were taken randomly from each sample.

### 2.11. Immunofluorescence Staining

Since the immunofluorescent punctate LC3 staining in the small PCT cells is difficult to detect, we used kidney epithelial COS7 and the lung A549 cancer cells to study Cd-induced K63-linked ubiquitination, SQSTM1 accumulation, and impaired autophagy flux. Cells seeded on glass coverslips were treated with Cd or bafA1, fixed with methanol, and indirect immunofluorescence was performed using antibodies against ubiquitinated proteins (FK2 1/200 or Ub 1/320), SQSTM1 (BD Transduction Laboratories™; #610833, 1:500), autophagy (LC3-II; clone 5F10; Nanotools, 1/500), and lysosomal markers (LAMP1, BD Pharmingen, 1 µg/mL, Clontech) to detect autophagosomes (LC3-II+) and autolysosomes (LC3-II+ and LAMP1+). To label mitochondria, 100 nM MitoTracker Red (molecular probes) was added to the medium for 25 min at 37 °C, followed by a quick wash in PBS and fixation. Nuclei were stained with ProLong Gold antifade reagent with DAPI (4′, 6-diamidino-2-phenylindole, Invitrogen). Pictures were taken with a 63× magnification lens using a confocal laser-scanning microscope (Zeiss LSM510 Meta, Marly le Roi, France) fitted with a 405, 488, 543, and 633 nm krypton/argon laser to detect simultaneous DAPI, fluorescein, rhodamine, and cyan5 chromophores. Alternatively, COS7 cells were transfected with a pH-sensitive mRFP-EGFP-LC3 reporter to distinguish autophagosomes showing both mRFP and EGFP signals from acidic autolysosomes that emit only an mRFP signal due to quenching of EGFP in acidic lysosomes.

### 2.12. Measurement of Protein Degradation

Degradation of long- and short-lived proteins was determined as previously reported [44,45]. Briefly, cells were incubated for 72 h at 37 °C in fresh DMEM/F12 medium containing 2% dialyzed FCS and 0.15 µCi of L-[^14^C] valine (Perkin Elmer, Courtaboeuf, France). Unincorporated radioisotope was removed by rinsing with DMEM three times. Cells were then chased with culture medium containing 10 mM cold valine. After overnight incubation, when short-lived proteins are degraded, the chase medium was replaced with fresh medium containing Cd. When required, NH_4_Cl (20 mM) or 3-methyladenine (3MA; 10 mM) were added to inhibit lysosomal and autophagic degradation, respectively. Protein degradation was then analyzed for 5 h to ensure optimal inhibition and to avoid toxicity. Cells and medium were harvested and precipitated with trichloroacetic acid (TCA) at a final concentration of 10% (*v*/*v*) at 4 °C. The samples were centrifuged (15,000× *g*, 10 min), and the acid-soluble radioactivity (released amino acid) was measured with a liquid scintillation counter. Acid-insoluble radioactivity (labeled proteins) was dissolved in 0.2 N NaOH and counted. The percentage of protein degradation was calculated by dividing the acid-soluble radioactivity recovered from both cells and medium by the sum of acid-soluble and acid-precipitable radio-activities. Non-lysosomal-dependent degradation (primarily mediated by proteasomes) was the percent of protein degradation resistant to NH_4_Cl. The contribution of autophagy was calculated by subtracting the radioactivity remaining after inhibition with 3MA from the total radioactivity. Alternatively, the cells were quickly chased for 2 min for both long- and short-lived proteins labeled. Short-lived protein degradation was then estimated by subtracting the radioactivity after a short chase (long- and short-lived proteins) from the radioactivity after a long chase (long-lived proteins). All experiments were performed at least six times with duplicate samples.

### 2.13. Quantitative RT-PCR Analysis

Total RNA was isolated from cells with TRIzol Reagent (Invitrogen), and quantitative RT-PCR analysis was performed using a SybrGreen PCR Master Mix (Applied Biosystems). The mRNA expression level of the NF-κB targets, *CXCL2* and *CCND1*, was quantified using the 2^[−ΔΔC(T)]^ method and normalized to the levels of the housekeeping gene (*RPLP0* or *TUBULIN*, Applied Biosystems). Relative gene expression changes are reported as the number of fold changes compared to the untreated cell samples (which were then set to 100).

### 2.14. NF-κB Luciferase Reporter Assay

To detect NF-κB-dependent gene transcription, PCT cells (0.8 × 10^6^/six-well dish) were transiently transfected using Fugene HD (Roche) with 1 µg of a firefly luciferase reporter gene controlled by a minimal thymidine kinase promoter, and 6 reiterated κB sites (pκBx6 TK-luc). The Renilla luciferase gene (0.01 µg) was co-transfected and used as an internal control. Following transfection, cells were allowed to recover for 24 h at 37 °C, depleted, and subsequently exposed for 8 h to Cd (5 μM) or LPS (5 µg/mL, used as positive control), alone or in combination with the IKK2 inhibitor (sc-514, 100 µM). The luciferase activities (firefly and Renilla) were measured 8 h later in the same sample using a dual-luciferase reporter assay (Promega) on a luminometer (Berthold, Germany). The firefly luciferase activity was adjusted by dividing the value by the corresponding Renilla luciferase activity. Each experiment was repeated four times. Data are expressed as means ± standard deviations of four replicates.

### 2.15. Electrophoretic Mobility Shift Assays

Total cellular extracts were prepared in lysis buffer (13 mM HEPES, pH 7.9, 350 mM NaCl, 20% glycerol, 1% NP-40, 1 mM MgCl_2_, 0.5 mM EDTA, 0.1 mM EGTA, 1 mM DTT, and protease inhibitors). After centrifugation (15,000× *g*, 15 min, 4 °C), supernatants were collected. NF-κB double-stranded probe (5′-GAT CCA AGG GAC TTT CCA TG-3′ of the Igk promoter’) was end-labeled with [γ-^32^P] ATP using T4 polynucleotide kinase and incubated with samples (10 µg) for 20 min at 30 °C. Cell extracts were preincubated with antibodies (2 µg) specific for p50/NF-κB1, p52, p65/RelA, RelB, c-Rel, or Bcl3 (Santa Cruz Biotechnology) for 30 min on ice before the addition of the labeled probe. DNA/protein and DNA–protein–antibody complexes were resolved on a 5% polyacrylamide gel in 0.5× TBE and detected by autoradiography.

### 2.16. Statistics

Results are given as means ± standard errors. Nonparametric Spearman correlation was applied to study correlations between different results, followed by Dunnett’s multiple comparison test, and data are presented as histograms. For all statistical analyses, significance levels were set at *p* < 0.05 (*).

## 3. Results

### 3.1. Cadmium-Induced Protein Ubiquitination was Independent of Oxidative Damage and Proteasome Impairment

Since Cd triggers massive oxidative damage (carbonylation) and subsequent impairment of proteasomes, the only reported mechanisms by which Cd induces ubiquitination, we tested this hypothesis [46,47,48]. Treatment of renal epithelial cells with 5 μM CdCl_2_ increased the levels of ubiquitin-conjugated proteins (Figure 2A). Ubiquitination was preceded by the depletion of cytosolic GSH (Figure 2B, left) and the generation of ROS, as indicated by the oxidation of the fluorescent probe H_2_DCFDA (Figure 2B, right) and the carbonylation of proteins (Figure 2C).

However, we found evidence that the Cd-induced ubiquitination and carbonylation were two independent responses: (i) the antioxidant vitamin E prevented ROS production and subsequent protein oxidation in Cd-treated cells (Figure 2B,C). But Cd was still able to induce ubiquitination in the presence of vitamin E (Figure 2C), confirming previous observations obtained with another antioxidant (ascorbic acid [49]). (ii) The addition of H_2_0_2_, a positive control for oxidative stress, was able to oxidize proteins but failed to induce protein ubiquitination per se (Figure 2C).

We then examined the contribution of impairment of proteasome on Cd-induced ubiquitination. Cd is thought to stabilize ubiquitinated proteins by inhibiting proteasomes, as suggested by the increased levels of cyclooxygenase 2, ATF5, NRF2, and Tp53 [50,51], four proteasome substrates. Nevertheless, Cd increases proteasomal degradation of eIF4E and Na/K ATPase [52,53]. We ruled out a role for proteasomal as a cause of Cd-induced ubiquitination in Cd-induced ubiquitination with three different approaches (Figure 2D,E). In vitro, whereas the chymotrypsin-like activity of the 20S proteasome was inhibited by MG132, we did not observe any proteasomal impairment associated with Cd treatment (Figure 2D, left). We then used metabolic labeling and pulse-chase experiments to measure the effects of Cd on cell proteolysis directly. We confirmed that the Cd inhibitory effect on both short- and long-lived proteolysis was independent of proteasomal inhibition, in agreement with [54] (Figure 2D, right). Consistently, the level of ubiquitinated proteins was further increased in Cd- and MG132-treated cells, indicating that the proteasome still degraded ubiquitinated proteins in Cd-treated cells. These observations do not provide evidence in favor of an essential role for oxidative damage and proteasomal shortage in Cd-induced ubiquitination.

### 3.2. Cd Is an Activator of K63-Linked Ubiquitination in Lung and Renal Cells

The nature of the Cd-induced polyUb chains remains elusive. We therefore used mass spectrometry to identify the specific Ub linkages. Cd-treated cells displayed accumulation of K48-linked Ub (K48-polyUb) and K63-polyUb chains, compared to the selective accumulation of K48-polyUb chains in MG132-treated cells (Figure 3A). Immunoblotting with an antibody that recognizes K63-polyUb chains confirmed dose-dependent K63-linked ubiquitination in Cd-treated cells, with a maximal effect at 5 µM (Figure 3B,C). In contrast, little, if any, K63-linked ubiquitination was detected with MG132 treatment, supporting the notion that K63-polyUb proteins are not degraded through proteasome (Figure 3B). Likewise, such extensive K63-linked ubiquitination was observed without enrichment in lung cancer (A549) and renal (PCT) epithelial cells, which are the primary routes for human exposure (Figure 3, Figure 4 and Figure 5), indicating that Cd-induced K63 ubiquitination is a significant phenomenon.

We then questioned how Cd increased ubiquitination. In addition to cadmium, many heavy metals are carcinogenic, such as lead, chromium, arsenic, and also the essential trace elements like cobalt on overload conditions [55]. Of metals, we found that K63-linked ubiquitination was also slightly enhanced by the addition of ZnCl_2_ but not by other heavy metals (Pb, Hg, Co, Fe) (Figure 3D). As Cd and Zn are in the same group of the chemical periodic table, it can be inferred from the so-called ionic mimicry concept that Cd may share with Zn the same targets [28,56]. In this regard, Cd is able to replace Zn and binds to ubiquitin [57,58], Zn finger domains, and the SH group within the active sites of ubiquitin ligases and deubiquitinases [59], a property that may help explain the dramatic upregulation of ubiquitination. However, the results of in vitro assays using free tetraubiquitin suggested that the addition of Cd (5 µM) had little, if any, direct effect on the activity of the ubiquitin ligases and deubiquitinases (Figure 3E,F).

### 3.3. Cd Selectively Impairs the Autophagy Degradation of Short-Lived Proteins without Interfering with the Autophagy Flux of Long-Lived Proteins

An alternative explanation for the accumulation of K63-linked ubiquitinated proteins besides the implication of proteasomes may concern defects in autophagy, a Zn-sensitive process that degrades K63-linked ubiquitinated cargos [5,9,32,33,34,35]. Inline, Cd induced higher molecular weight (Hmw) forms of Beclin 1/BECN1, an essential component of PtdIns3K complexes that mediates the formation of autophagosomes. By contrast, the inhibition of the proteasome by MG132, which caused a specific increase in K48-linked ubiquitin chains (Figure 3A), did not induce any difference in the BECN1 pattern (Figure 4A). This suggested that higher MW of BECN1, likely ubiquitinated forms, did not depend on the K48 but instead on K63-linkage. Of interest, when BECN1 is ubiquitinated by K63-linkage, BECN1 is released to activate the PI3P kinase activity and thereby initiate autophagy [60].

We therefore performed biochemical, ultrastructural, and functional assays to analyze the activity of the autophagy pathway in Cd-treated kidney epithelial and lung cancer cells (Figure 4, Figure 5 and Figure 6). We found that the previously reported autophagic response of Cd-treated cells did not result from enhanced proteolysis but rather from a biphasic autophagic response, in which initial engagement in autophagy was followed by retarded compromised autophagic degradation [29]. Cd (5 µM) induced a marked and sustained conversion of LC3-I to LC3-II that lasted 8 h in PTC cells (Figure 4B). The LC3-II levels were further increased after 2 h of Cd treatment when the lysosomal protease inhibitors (E64d and pepstatin A, Figure 4C, left) were added, suggesting that the flux in autophagy was functional. At this early time point (2 h) of treatment, transfection of a pH-sensitive, mRFP-EGFP-LC3 reporter allowed us to confirm the efficient formation of autophagosomes (yellow, mRFP-EGFP, and LAMP1-negative; arrowhead) and acidic autolysosomes (only mRFP, and LAMP1-positive; arrow, Figure 4D), arguing against a Cd-induced defect on the v-ATPase activity [61], which is in agreement with a previous report [62]. As for HBSS, Cd-treated cells (2 h) exhibited only a few green dots and a predominant red LC3 staining that was indistinguishable from purple LAMP1 labeling, supporting an autophagic flux to acidic autolysosomes. However, at 6 h with Cd, the autophagic flux was severely impaired as no difference in the amount of LC3-II was observed in the presence of E64d and pepstatin A (Figure 4C, right). Autophagosomal maturation was consistently defective, as shown by a marked accumulation of yellow LC3 (yellow, colocalized red mRFP and green EGFP fluorescence, and LAMP1-negative) (Figure 4D).

Under these conditions, Cd did not compromise degradation of long-lived proteins by autophagy (Figure 4E), which is in agreement with a previous report [63]. Consequently, we assumed that the Cd-induced defect did not disrupt the global flux of autophagy but rather a specific defect in terms of substrate selectivity. Previous studies have demonstrated that long-lived proteins are primarily degraded by starvation-induced autophagy [11]. By inference, it is generally thought that selective and constitutive autophagy degrades long-lived proteins in a similar manner. When the cells were chased for 2 min to label both long- and short-lived proteins, Cd impaired the autophagy clearance of short-lived proteins without interfering with the flux of autophagy of long-lived proteins (Figure 4E).

### 3.4. Inhibition of the Degradation of Three Selective Autophagy Substrates—SQSTM1, Ubiquitinated Proteins, and Mitochondria by Cd

From the above results, we anticipated that the defect in autophagy caused by Cd might accumulate three substrates that are degraded by selective autophagy: SQSTM1, ubiquitinated proteins, and mitochondria. Of these, we focus on Sequestosome 1 (SQSTM1/p62), a multifunctional protein that interestingly contains a Zn binding (ZZ) domain that could bind to Cd [37,38]. This protein also functions as a critical selective autophagy receptor that binds to LC3 and K63-ubiquitinated proteins, delivering them for degradation by autophagy [13]. Meanwhile, SQSTM1 is degraded by autophagy. Thus, SQSTM1 degradation is a marker of autophagic flux, along with LC3-II [11]. Surprisingly, we noted a high expression of SQSTM1 that did not increase further when flux was inhibited with Cd and E64d/peps at 2 h, in contrast to LC3-II. This contradictory result suggests that SQSTM1 could be stabilized before the autophagy flux was impaired (Figure 4C). Strikingly, SQSTM1 was detected both in the detergent-soluble and -insoluble fractions of control cells (Figure 5A). In contrast, the apparent loss of SQSTM1 in the soluble fractions upon Cd-treatment was due to its increased sequestration into insoluble cellular aggregates (Figure 5A, left panels). Following Cd treatment, a ladder of higher molecular weight molecules of SQSTM1 was observed only in the insoluble fraction, likely reflecting SQSTM1 ubiquitination and aggregation [36]. Likewise, Cd induced a similar accumulation of K63-linked ubiquitinated proteins and LC3-II within the aggregates (Figure 5A, middle and right panels).

A long-standing hypothesis assumes that protein aggregates may sequester and protect the injured cell from harmful misfolded and damaged proteins. The oxidative stress was demonstrated to drive aggregate formation through increased SQSTM1 transcription, protein ubiquitination, and subsequent recruitment of LC3 to aggregates [64]. However, by adding vitamin E, we observed that the accumulation of SQSTM1, the K63-linked ubiquitination, and the conversion of LC3-II in Cd-treated cells were independent of oxidative stress (Figure 5A). At this stage, it was of interest to address whether SQSTM1, LC3-II, and UB may colocalize in Cd-treated cells. Immunofluorescence analysis showed that the ubiquitinated proteins, SQSTM1, and LC3-II were present at low levels in untreated cells (Figure 5B, inset). Numerous ubiquitin- (Figure 5B, magenta) and SQSTM1 (Figure 5B, blue)-positive puncta increased in both number and size when cells were treated by Cd. The merged image demonstrates nearly complete colocalization in the ubiquitin and SQSTM1 spots (purple), regardless of their size. By contrast, the larger ubiquitin- and SQSTM1-positive inclusions were also positive for LC3-II (Figure 5B, white). Furthermore, we observed that the aggregates were dispersed throughout the cytosol of Cd-treated cells, in sharp contrast to juxtanuclear aggresomes reported for misfolded proteins [65]. Similarly, Cd led to disruption of the mitochondrial network (labeled with MitoTracker dye, magenta), which was concentrated at the site of SQSTM1 aggregates (Figure 6A,B, blue), consistent with the fact that SQSTM1 is an autophagy receptor for mitochondria.

SQSTM1, ubiquitinated proteins, and mitochondria clustered at locations that could not be distinguished from LC3-positive dots by confocal microscopy (Figure 5B and Figure 6B, green). These results suggest that these three substrates were recruited to autophagic vesicles. To elucidate the step(s) in the autophagy pathway that Cd may disrupt, we performed ultrastructural analyses (Figure 5C and Figure 6C). High-magnification microscopy of Cd-treated cells showed the formation of electron-dense inclusion bodies, likely aggregates of ubiquitinated proteins (arrowheads). At 2 h of Cd treatment, the association of ubiquitinated proteins within autophagic vesicles was confirmed: small electron-dense aggregates (arrowheads) were efficiently trapped within autophagic vesicles that also contained small electron-lucent double-membrane, multivesicular, and lamellar vesicles (Figure 5C, arrows, left). However, after 6 h, when the ubiquitinated proteins accumulated, the size of the inclusion bodies increased (Figure 5C and Figure 6C). Double-membrane vesicles, membrane remnants, and swollen mitochondria with a reduced number of cristae often appeared to be trapped at the periphery of these large inclusions (Figure 5C and Figure 6C). While the vesicles formed at early Cd-treatment times were all typical autophagic vesicles with a recognizable content (Figure 5C, right), all of the vesicles clustering at the periphery of the inclusion bodies at later times were still double-membraned but appeared to be “empty” (Figure 5C and Figure 6C, right), even if they were close to proteins and mitochondria. It may be possible that the vesicles trapped in the Cd-induced protein aggregates were stalled autophagic intermediates. Detection by immunoblotting of the LC3-II autophagosomal protein exclusively in the detergent-insoluble fraction of Cd-treated cells supports this possibility (Figure 5A, right). Accordingly, Cd treatment in PCT and A549 cells faithfully recapitulated the same response with regard to the K63-linked ubiquitination, SQSTM1 aggregation, and sustained LC3-II conversion (*data not shown,* Chargui A. IRCAN, Nice, France, 2021). Collectively, these observations suggest that the formation of inclusion bodies hampered the biogenesis or maturation of autophagic vesicles.

In agreement with the above-mentioned Cd inhibition of constitutive autophagy (Figure 4E), we observed selective accumulation of K63-ubiquitinated proteins under Cd and serum/nutrient-rich conditions (constitutive) but not in starved conditions (Figure 7A). As a proof-of-concept, comprehensive analysis of the pharmacological inhibition of autophagy confirmed that constitutive- and starvation-induced autophagies targeted different types of cargo. Inhibition of autophagy degradation with bafA1 or CQ treatment also drove accumulation of K63-linked ubiquitinated proteins, SQSTM1, and LC3-II within detergent-insoluble aggregates under serum- and nutrient-rich conditions (constitutive) but not in starved conditions (HBSS; Figure 7A–C, Appendix A). Likewise, bafA1 treatment led to the accumulation of ubiquitinated proteins into autolysosomes under nutrient-rich conditions, while it had no effect when cells were starved (Appendix A). This supports the notion that ubiquitinated proteins are selective targets for constitutive autophagy. However, bafA1 treatment inhibited degradation by autophagy under both conditions, as observed at the ultrastructural level by the accumulation of single-membrane autolysosomes with undigested electron-dense bodies (Appendix A, inset). We then confirmed that constitutive autophagy selectively degraded primarily short-lived proteins, while starvation-induced autophagy degraded primarily long-lived proteins, in agreement with [11] (Figure 7D). Of note, bafA1 inhibited the degradation of both long- and short-lived proteins, in contrast to Cd. Our data therefore reveal that the carcinogen Cd is a unique inhibitor of selective autophagy.

### 3.5. Cadmium Induces Sustained Activation of the NF-κB Pathway

Critical for carcinogenesis, we then explored whether Cd could activate a signaling pathway regulated by K63 ubiquitination, SQSTM1, and autophagy. One candidate was the NF-κB pathway, which is well-known to be induced by Cd (for review, see [66]), but the underlying mechanism remains elusive.

As an initiating event, we then looked for a partner of SQSTM1, the K63 deubiquitinase CYLD, which is a tumor suppressor that constitutively regulates the NF-κB pathway by removing K63-linked ubiquitin chains from TRAF6 [10]. We found that Cd lowered the cytosolic levels of soluble CYLD proteins and led to its accumulation in detergent-insoluble fractions (Figure 8A, left). In addition to aggregation, Cd induced CYLD inactivation, as shown by an increase in high molecular weight (likely ubiquitinated) forms of CYLD and the cleavage of CYLD into several inactive fragments of ~72, 50, and 40 kDa [67], two events that were not recapitulated by proteasomal inhibition (*data not shown*, Chargui A. IRCAN, Nice, France, 2021). These results indicate that Cd reduced CYLD availability and likely its K63 DUB function.

As a readout of CYLD inactivation, Cd triggered the accumulation of high molecular weight (likely ubiquitinated) forms of K63-Ub ligase TRAF6, a marker of activation [68] (Figure 8A, right). Analysis of WCL by anti-K63 polyUb immunoblotting revealed that Cd induced at 30 min K63 ubiquitination of several cellular proteins of 170, 120, 80, 57, and 40 kDa, a pattern similar to that observed at 30 min in LPS-stimulated PCT cells (Figure 8B). Moreover, ubiquitination of cellular substrates started at 30 min of Cd treatment and lasted for 8 h. Most of the ubiquitinated proteins aggregated into detergent-insoluble bodies in Cd-treated cells (Figure 8C). This robust and sustained ubiquitination distinguished the carcinogen Cd from LPS, and the other cytokines and growth factors so far studied.

Downstream of K63 ubiquitination, we first assessed NF-κB activation by quantifying increases in phosphorylation of IκBα. Cd induced sustained and hyper-phosphorylation of IκBα that started at 1 h and lasted for 8 h (Figure 8D). In contrast, LPS resulted in transient IκBα phosphorylation at 0.5 h that returned to baseline levels after 2 h. Interestingly, in Cd-treated cells, we detected a high molecular weight form of phosphorylated IκBα, which was likely phosphorylated and ubiquitinated (Figure 8D, upper). These findings suggested that IκBα might undergo phosphorylation, ubiquitination, but not degradation in Cd-treated cells. In this regard, the recent observation that phosphorylated IκBα is a selective autophagy substrate is of great interest [69]. The phosphorylation and ubiquitination of IκBα appeared to depend on IKKβ as both were blocked by a selective IKK2 inhibitor, sc-514 (Figure 8K). Therefore, in contrast to LPS, Cd resulted in sustained activation of the NF-κB signaling pathway in PCT cells.

The timing of IKK activation correlated with the time-course of NF-κB activation observed in the gel shift assays (Figure 8E). Cd-induced activation of NF-κB occurred within 1 h and was sustained for at least 4 h (Figure 8E, top). Two Cd-stimulated NF-κB-binding complexes were detected in nuclear extracts: a minor slower migrating band, and a major band, which migrated with higher mobility (Figure 8E). Electrophoretic mobility super-shift analyses revealed that p65, and to lesser levels p50, p52, cRel, and Bcl3, antibodies retarded the migration of the complex band, indicating that p65 heterodimers were induced in response to Cd (Figure 8F). These results were confirmed by the transfection of an NF-κB-driven reporter (Figure 8G) and the downstream robust mRNA expression of the inflammatory chemokine *CXCL2* (10-fold of control value; Figure 8H).

We also measured the levels of *CCND1* (Cyclin D1), another NF-κB-responsive gene, a marker of NF-κB activation. Cd induced strong activation of the *CCND1* promoter (Figure 8I) and the expression of the CCND1 protein (Figure 8J), with a time course that paralleled Cd-induced NF-κB activation. Treatment with the IKK inhibitor blocked Cd-induced NF-κB activation, –*CXCL2*, and CCND1 expression (Figure 8H,K). ROS were again not involved in the Cd-induced NF-κB pathway from K63 ubiquitination, IκBα phosphorylation, to downstream CCND1 and *CXCL2* expression (Figure 8H,K). Similar sustained activation of the NF-κB pathway was observed in Cd-treated A549 lung cancer cells (*data not shown*, Chargui A, IRCAN, Nice, France, 2021). This demonstrated that low doses of Cd activated K63-linked ubiquitin-dependent NF-κB signaling, leading to inflammation and cell proliferation.

## 4. Discussion

Although the tumor-suppressive function of autophagy was first described in 2003 [14,15], the roles of constitutive/selective autophagy and inducible/bulk autophagy in cancer remain poorly understood. Indeed, the most serious bottleneck in autophagy research is the lack of pharmacological inhibitors and specific genetic tools for the manipulation of these types of autophagy [11]. Therefore, a key feature of our strategy was to inhibit the autophagy pathway with a widespread carcinogen, cadmium, to accumulate autophagy substrates that may be required for tumor initiation.

Even though Cd is a well-recognized carcinogen, its underlying molecular mechanism is far from understood, particularly at a low-level of environmental exposure. Indeed, most studies carried out so far with cellular and animal models have focused on the induction of severe genotoxic effects (mutation, apoptosis, or necrosis), albeit only following high toxic Cd concentrations (0.1–10 mM, [70]). As a result, the nongenotoxic effects of Cd and, more generally, of carcinogens at environmental levels are not well-understood. The concentrations of Cd of 5–20 μM used in this study were within the range documented for lung and kidney tissues in humans exposed to low environmental doses (µg/g [71]).

Our study reveals novel and critical consequences of Cd exposure at such subtoxic doses: Cd is the first example of a carcinogen that induces K63-linked ubiquitination (Figure 3), in part by inhibiting autophagy (Figure 4, Figure 5 and Figure 6). Specifically, our analysis of Cd-compromised autophagy challenges the “quantitative” model in which basal and inducible autophagies degrade the same long-lived proteins (Figure 4). Instead, we provide evidence of differences in the nature of the cargo targeted by each type of autophagy (Figure 4, Figure 5, Figure 6 and Figure 7). Both Cd and bafA1 induced the accumulation of three substrates selectively degraded by autophagy, i.e., K63-linked ubiquitinated proteins (Figure 3, Figure 5 and Figure 7), SQSTM1 (Figure 5 and Figure 7), and mitochondria (Figure 6), which occurred only under nutrient-rich conditions. Failure to eliminate these selective cargos was not due to alterations in the “general” autophagy process, as degradation of long-lived proteins occurred (Figure 4E and Figure 7D). While more studies are needed to explore the underlying mechanisms, our data support a “qualitative” model in which the “selective” and “non-selective” autophagy pathways are mutually exclusive regarding their substrates: basal autophagy ensures the selective clearance primarily of ubiquitinated short-lived cargoes that switches to bulk degradation of non-ubiquitinated long-lived proteins upon nutrient starvation.

The specific disruption of constitutive/selective autophagy with Cd may be a key event in carcinogenesis. In support of this hypothesis, malignant cells frequently display a lower autophagic activity than their normal counterparts. Many autophagy machinery components are downregulated or mutated in many cancers (melanoma, myeloma, glioma, colon, and lung cancers) (for reviews, see [12,16]).

Therefore, how could Cd specifically inhibit selective autophagy and not non-selective autophagy? Thus far, dogma dictates that the oxidative stress generated by this metal (Figure 2) damages proteins and lipids [46]. As a quality control process, autophagy is then stimulated to dispose of the damaged cellular components [12]. Thus, it is tempting to speculate that excessive ROS production may inhibit the autophagy flux by the oxidation of the autophagy machinery [72] and the peroxidation of autophagic membranes [73]. Yet, ROS are produced by many stimuli that trigger both selective autophagy and non-selective autophagy. Likewise, ROS are signaling mediators that randomly target the autophagy machinery and membranes [64]. Common to both selective autophagy and non-selective autophagy, excessive ROS production may thus shut down both pathways. We demonstrated that even though Cd induced ROS production and protein oxidation (Figure 2B,C), addition of two ROS scavengers (NAC or vitamin E) failed to reduce the K63 ubiquitination (Figure 2C and Figure 5A), accumulation of SQSTM1, formation of aggregates, and lipidation of LC3-II (Figure 5A) induced by Cd. This suggests that Cd-induced responses (K63 ubiquitination and selective autophagy) were regulated independently of ROS production.

One clue may come from comparison with the other heavy metals (Figure 3D). In addition to Cd, many metals are carcinogenic to humans, such as lead, chromium (VI), arsenic, and the essential trace elements like cobalt on conditions of overload [55]. Among the nonessential (Cd, Pb, and Hg) and essential trace elements (Co, Zn, and Fe) studied, we found that Cd was the most potent inducer of K63 ubiquitination. Only one essential element, Zn, was also able to induce K63 ubiquitination (Figure 3D). Of interest, both Cd and Zn are group IIb metals that share a common affinity for sulfhydryl groups and likely the same molecular targets. By displacing Zn, one might assume from the so-called ionic mimicry concept that Cd would inhibit the Zn-dependent enzymes, disturb folding of Zn-domain, and interfere with Zn-regulated physiological processes [28,56,57,58,59].

Approximately 3200 proteins (10% of the human proteome) have Zn-finger regions or Zn domains and require Zn to function correctly [74]. To protect the cell from Cd toxicity, we assume that Cd-bound proteins can be recruited into aggregates and autophagosomes. This is in line with the emerging role of autophagy as a defense response to metals [32,33,34,35]. Cd could then bind a zinc-specific adaptor in a mechanism akin to selective autophagy. A key enriched candidate within these Cd aggregates is SQSTM1, a scaffold signaling protein (Figure 5), the accumulation of which is essential for the development of lung and renal carcinoma [75,76,77]. At the molecular level, SQSTM1 is also a selective autophagy receptor that self-assembles, and binds K63-linked ubiquitinated proteins and LC3 for their selective degradation through autophagy [36]. It should be stressed that SQSTM1 contains a ZZ-type Zn-finger (ZZ) domain, in particular the Cys-X2-Cys residues, which are essential for SQSTM1 aggregation and autophagosome formation [13,37]. Of note, several other sequestosome-related genes such as *sqst-3* and *sqst-5* are associated with resistance to high levels of exogenous Zn or Cd [78,79]. Therefore, it is possible that the binding of the SQSTM1 ZZ domain to Cd could stabilize SQSTM1 from autophagy by disrupting its folding. The addition of Cd would then commit to SQSTM1-aggregation (Figure 5) and activation of SQSTM1-dependent signaling (Figure 8).

Critical for carcinogenesis, Cd hijacks key pathways that control normal development. We focused our attention on the NF-κB pathway because of several arguments: (i) The cancer cells are “addicted” to this signaling, which drives oncogenesis, inflammation, cancer progression, and resistance to therapy. (ii) Several seminal studies, both in vitro and in vivo, demonstrated that SQSTM1 is essential for NF-κB signaling, and thereby for cell transformation, lung, and renal cancer development [75,76,77]. (iii) Likewise, the NF-κB pathway is regulated by K63 ubiquitination and selective autophagy [7,80]. Among SQSTM1 signaling partners, we provide the first evidence that the activity of CYLD K63 deubiquitinase, a tumor suppressor, was repressed by Cd, likely via ubiquitination, processing, and aggregation (Figure 8A). Downstream, the inhibition of CYLD would release the K63 ubiquitin ligase TRAF6, another SQSTM1 partner (Figure 8A). Of interest, the same K63 ubiquitin ligase TRAF6, which triggers NF-κB activation in response to cytokines, also ubiquitinates ULK1 [81] and BECN1 [60], promoting the formation of autophagosomes. These autophagosomes turn out not to be formed for bulk degradation but selective autophagy/xenophagy as part of the host immune responses. Similarly, we observed that Cd targeted the activation of the TRAF6/NF-κB pathway (Figure 8), the downstream ubiquitination of BECN1 (Figure 4A), and the formation of autophagosomes that selectively degraded short-lived proteins (Figure 4). Such an intricate feedback loop between K63 ubiquitin signaling and selective autophagy may allow the cell to fine-tune the NF-κB program to a transient inflammatory signal. However, the accumulation of these ubiquitin linkages in response to Cd may overwhelm selective autophagy, amplifying the NF-κB pathway. As a result, Cd induces the persistent release of pro-inflammatory cytokines (CxCl2/MIP-2, TNFα, IL1α, IL1β, IL6, and IL8) that facilitate crosstalk with infiltrating immune cells and thereby create a pro-carcinogenic inflammatory environment [82].

Besides signaling, it is well-known that Cd damages mitochondria by “ionic mimicry,” which utilizes iron and calcium transporters and disrupts the electron transport chain [83]. We show here that Cd triggered the recruitment of damaged mitochondria within K63-ubiquitinated aggregates. In keeping with this observation, SQSTM1 was recently found to translocate to and induce clustering of damaged mitochondria as an ultimate cell defense before harm occurs [84]. Together, these observations suggest a new carcinogenic mechanism in which the sequestration of depolarized mitochondria within this K63-ubiquitinated cargo may protect the Cd-exposed cell from oxidative stress and apoptosis. Concomitantly, Cd-induced mitochondria sequestration may rewire the Warburg effect from normal oxidative phosphorylation to glycolytic metabolism of the tumor [85].

## 5. Conclusions

Our results collectively suggest a new model in which there are two autophagy types: constitutive/selective autophagy and starvation/bulk-induced autophagy, which degrade different cargoes and fulfill distinct roles in cancer. In this scenario, only constitutive autophagy that keeps K63 ubiquitin signaling in check is expected to prevent tumor initiation. Its disruption by Cd likely plays critical roles in all of the stages of carcinogenicity (Figure 1). At heart, the formation of protein aggregates, generally associated with neurodegenerative diseases, turns out to play a significant role in carcinogenesis by the sequestration and inactivation of two tumor suppressors, CYLD and selective autophagy, as previously reported for Tp53 family members [86,87]. Although we were unable to identify the initiating event, several lines of evidence suggest that selective autophagy may be a Zn-regulated process [32,33,34,35], in part because its selective cargo receptor SQSTM1 could bind Zn in a domain that links K63 Ub and LC3, and thus autophagosome formation [13,37]. Therefore, we propose that Cd ionic storm may displace Zn from the SQSTM1-dependent selective autophagy. This Cd hijacking may create a vicious cycle that amplifies K63-linked ubiquitination, protein aggregation, and downstream activation of the NF-κB transcription program, providing proinflammatory, proliferative, and survival advantages to the initiating tumor cells.

For future study, a deeper understanding of Cd carcinogenesis will require a more detailed identification of K63-ubiquitinated targets trapped within SQSTM1 aggregates using large-scale proteomics. In particular, it will be interesting to determine whether other components of the NF-kB pathway could be similarly accumulated in the aggresome, such as the K63 DUB A20, the kinases IKKA, IKKB, and IKKG, the transcription factors RELA, and its inhibitor IκBα, which are degraded by selective autophagy [19,80]. Among the attractive short-lived candidates, we were interested in IκBα, an inhibitor that keeps the transcription factor NF-κB inactive in the cytosol. In response to inflammatory cytokines, IκB is phosphorylated, ubiquitinated, and degraded, thereby allowing NF-κB to translocate to the nucleus. We show here that Cd induced the accumulation of high molecular weight forms of phosphorylated IκBα, while selective autophagy was inhibited, suggesting that ubiquitinated P-IκBα relied on selective autophagy for degradation (Figure 8D). Along this line, blocking autophagy with CQ was recently reported to stabilize the interaction of SQSTM1 with ubiquitinated phospho-IκBα and prevent their selective degradation [69]. In keeping with our results, the Mallory-Denk-bodies, which share with Cd aggregates, SQSTM1, and ubiquitin, also contain ubiquitinated phospho-IκBα. Importantly, this IκB-sequestration into Mallory bodies was demonstrated to release and activate NF-κB [88]. Therefore, this raises the question of a novel model in which impaired SIGNALphagy and protein aggregation may account for persistent NF-κB activation and downstream severe inflammatory and proliferative responses in Cd-exposed cells. A better understanding of the molecular balance between signal integration (SQSTM1 aggregates) and termination (SQSTM1-dependent SIGNALphagy) and its disruption by Cd will offer new insights into diseases showing aggregation and therapeutic avenues aimed at specifically restoring selective autophagy.

## Figures and Tables

**Figure 1 cancers-13-02490-f001:**
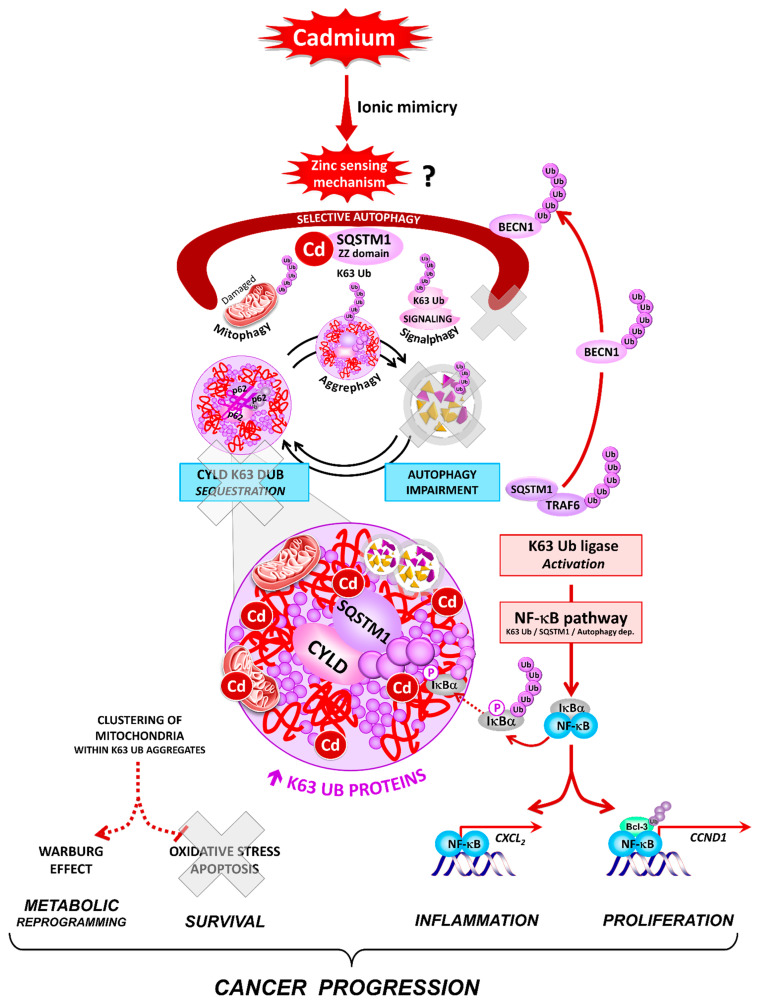
Activation of K63 ubiquitination and subversion of selective autophagy in Cd-carcinogenesis. Arrows or interactions do not imply a direct but rather a deduced or presumed (dashed lines) order of different signaling components.

**Figure 2 cancers-13-02490-f002:**
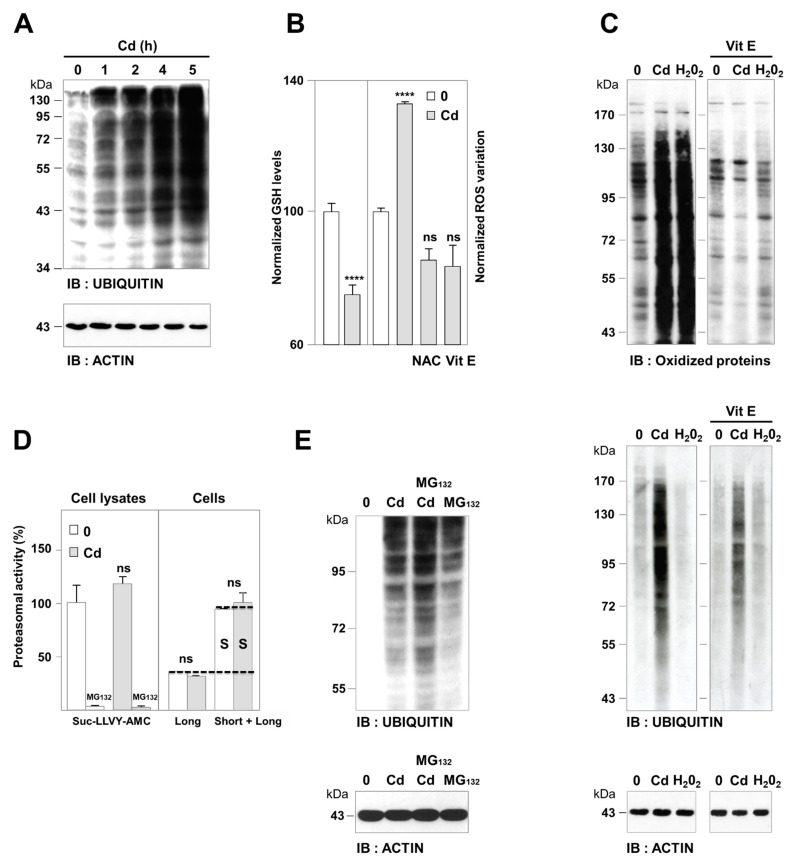
Cd induced protein ubiquitination independently of oxidative damage and proteasome impairment. (**A**) Accumulation of ubiquitinated proteins. PCT cells were incubated in the absence (0) or the presence of Cd (5 µM). Ubiquitinated proteins were detected by immunoblotting (IB) of whole-cell lysates (WCL) with anti-ubiquitin antibodies. (**B**) ROS production. The intracellular levels of GSH (left) and ROS (carboxy-H_2_DCFDA, right) were measured after treatment with Cd (5 μM, 30 min). Cells were treated with Cd in combination with two antioxidants: NAC (10 mM, also a scavenger of Cd) or vitamin E (100 μM, Vit E), where indicated. The results are expressed relative to the level of untreated cells, which was given an arbitrary value of 100. Not significant (ns), **** *p* < 0.0001. (**C**) Detection of oxidized proteins in WCL using the OxyBlot kit. The positive controls, cells stimulated by H_2_0_2_ (1 µM), showed high protein carbonylation that was completely inhibited by Vit E. (**D**) Proteasomal activity. Cells were treated with Cd (5 µM, 5 h), lysed, and the proteasomal activity was measured either in vitro from the degradation of specific Suc-LLVY-AMC peptide (right) or in live cells from the degradation of short-lived (S) and long-lived proteins (left, proteasomal non-lysosomal proteolysis, NH_4_Cl-resistant), and (**E**) the accumulation of ubiquitinated proteins (immunoblotting). The positions of the molecular weight markers are indicated on the left in kDa. See Appendix A for cell viability and Appendix A for uncropped images of immunoblots.

**Figure 3 cancers-13-02490-f003:**
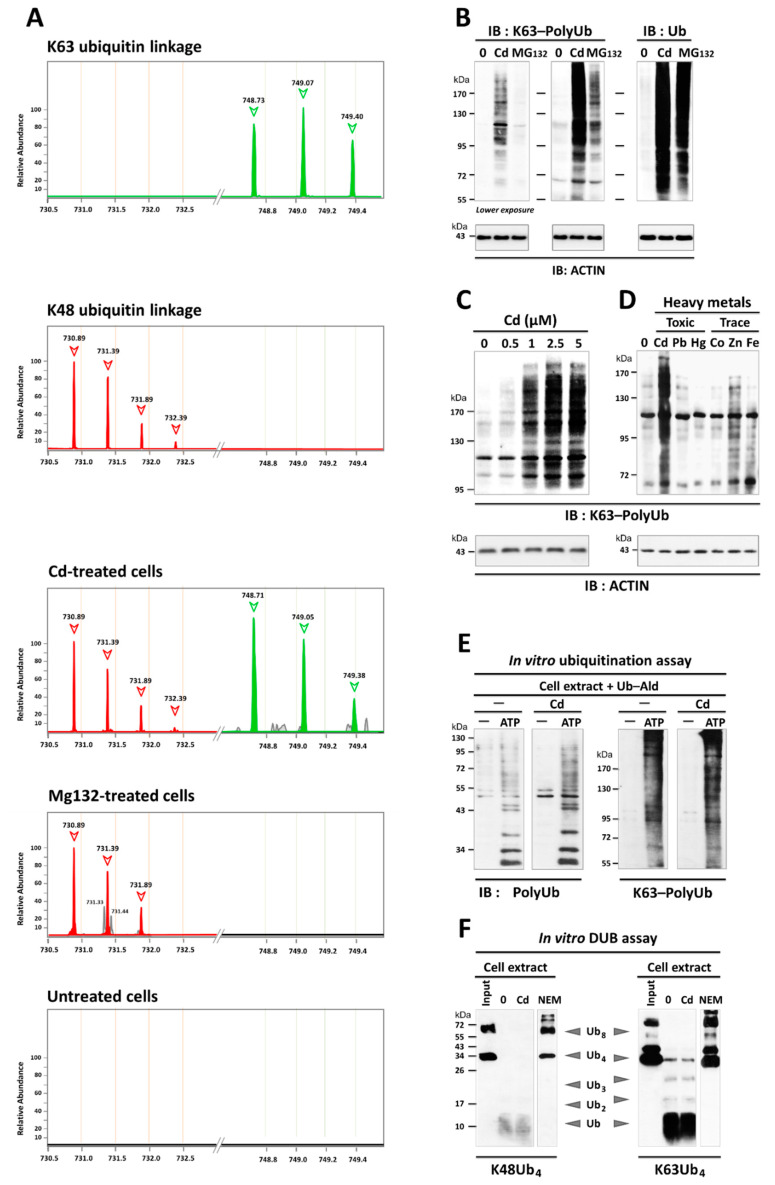
Enhanced K63-linked ubiquitination by Cd. (**A**) Identification of the ubiquitin linkage by mass spectrometry. Insoluble fractions of Cd- and MG132-treated cells (as control of proteasomal inhibition) were digested with trypsin. The resulting peptides were analyzed by MS. The important peaks corresponding to K63- and K48-Ub linkages are indicated in green and red with their ideal masses (m/z, arrows), respectively. (**B**). Accumulation of K63-linked ubiquitination in response to Cd, but not to MG132 (10 µM). Note that little, if any, K63-linked ubiquitination was detected with MG132 (only after overexposure). (**C**) Dose–response effect of Cd on the levels of K63-linked ubiquitination. Note that at 5 µM, Cd could achieve K63-linked ubiquitination while it did not adversely affect cell viability (Appendix A). (**D**) Comparison of the K63-linked ubiquitination levels induced by heavy metals. PCT cells were incubated for 5 h in the presence of toxic (Cd (5 µM), Pb (NO_3_)2 (5 µM), HgCl_2_ (5 µM)) and trace metals (CoCl_2_ (100 µM), Zn(NO_3_)_2_ (100 µM), or FeCl_2_ (100 µM)). Ubiquitinated proteins were then detected by WCL immunoblotting using pan ubiquitin (positive control of proteasome inhibition) or K63-polyUb-specific antibodies. (**E**) Cd did not increase protein ubiquitination in vitro. Lysates from untreated PCT cells (containing E1, E2, and E3 ubiquitin ligases) were incubated with ubiquitin and ATP in the absence or presence of Cd (5 µM). After incubation at 37 °C for 16 h, the products were subjected to immunoblotting using either pan polyUb (FK2) or K63-polyUb antibodies. No ubiquitination was observed in the absence of ATP. (**F**) Cd did not directly inhibit the K48 DUB and K63 DUB activities in vitro. Cell extracts (as a source of DUB) were preincubated with Cd (5 µM) or NEM (DUB inhibitor, 10 mM) before the addition of 0.25 µg tetraubiquitin K48Ub_4_ or K63Ub_4_ as substrates. After incubation at 30 °C for 16 h, the reaction was analyzed by immunoblotting with anti-ubiquitin. Input tetraubiquitins are shown. See also Appendix A for uncropped images of immunoblots from Figure 3B–D.

**Figure 4 cancers-13-02490-f004:**
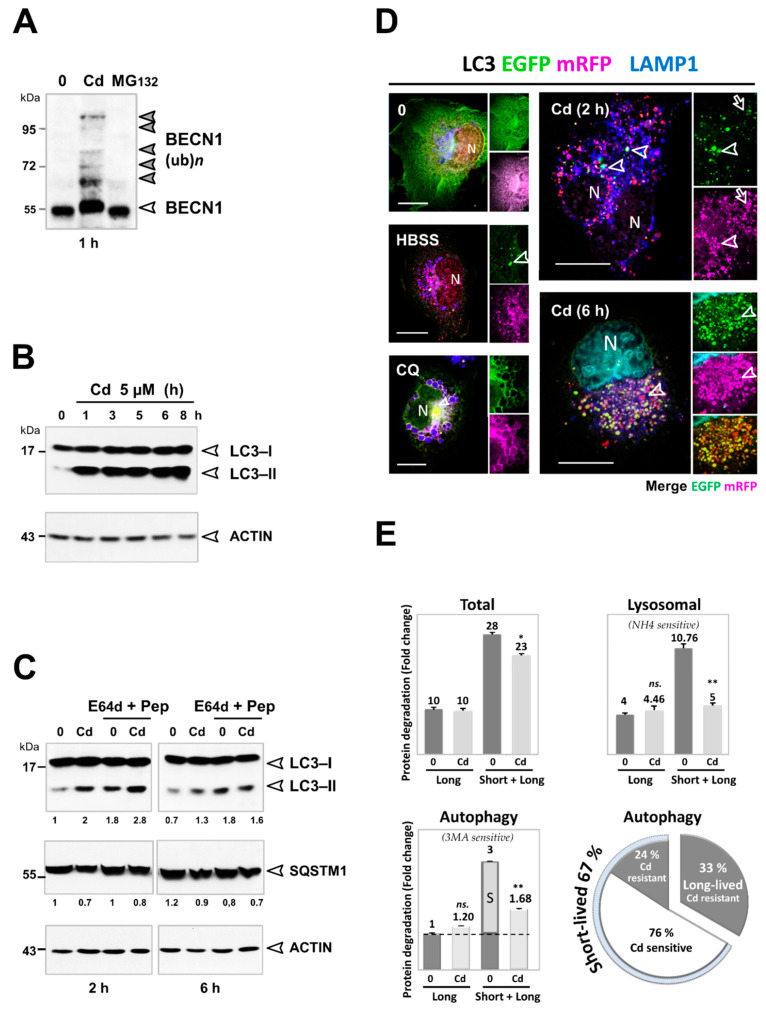
Cd-induced accumulation of autophagic vesicles correlates with a defect in autophagic degradation. (**A**) Ubiquitination of BECN1 in response to Cd, but not to MG132 (10 µM, 5 h). High molecular weight forms of BECN1, likely BECN1 polyubiquitination (grey arrowheads), were detected by immunoblotting of Cd-treated cell lysates. ACTIN was used as a loading control (see Figure 3B). (**B**) Time course of LC3-II conversion induced by Cd. (**C**) The autophagy flux of Cd-treated cells was analyzed by conversion of LC3-I to LC3-II (immunoblotting), in the absence or the presence of lysosomal protease inhibitors (E64d and pepstatin A; E64d + Pep; 10 µg/mL). The positions of LC3-I and LC3-II are indicated. Numbers under each lane are the pixel intensities of the LC3-II and SQSTM1 lanes (quantified using Image J software and normalized to the untreated lane). (**D**) Cells were transfected with the pH-sensitive, tandem mRFP-EGFP-LC3 reporter, treated with Cd (5 µM 2 and 6 h), HBSS (6 h), or chloroquine (CQ, 100 µM, 6 h), and labeled with anti-LAMP1, a specific lysosomal marker. This allowed us to clearly distinguish between neutral autophagosomes displaying both green and red fluorescence (yellow, LAMP1-negative; arrowhead) and acidic autolysosomes with red (pseudo-colored in magenta) and blue fluorescence (purple, LAMP1-positive, mRFP-LC3, because of quenching of EGFP in the acidic lysosomal environment; arrow). In CQ-treated cells (a weak base), all green spots were also red (magenta), and the label of LAMP1 corresponded to neutral autolysosomes. Twenty to twenty-five IF pictures were taken randomly from each sample. All cells displayed these modifications. The most representative is presented in the Figure. (**E**) Autophagic degradation of short-lived proteins (S) was selectively inhibited by Cd. PCT cells were metabolically labeled with [^14^C] valine and chased with 10 mM cold valine for either 2 min (to label both short- and long-lived proteins, “Short + Long”) or 16 h (only labeled long-lived proteins, “Long”). After labeling and chase, cells were treated with Cd (5 µM) in complete serum- and amino acid-supplemented media for 5 h. The level of protein degradation in response to Cd was then measured by the release of trichloroacetic acid-soluble [^14^C] valine from cells, as described in the Materials and Methods Section. Protein degradation by macroautophagy (which was given the arbitrary value of 1) was calculated from the difference between the [^14^C] valine released from cells treated without (Total) and with a specific inhibitor of macroautophagy, 3-methyladenine (3MA, 10 mM). 20 mM NH_4_Cl was added to cells to obtain lysosomal (difference between total- and the NH_4_Cl-resistant-proteolysis). Each result is presented as a fold change relative to the autophagy degradation in untreated cells (mean ± duplicates). Data are representative of three independent experiments. See Appendix A for uncropped images of immunoblots. Not significant (ns), * *p* < 0.01, ** *p* < 0.005.

**Figure 5 cancers-13-02490-f005:**
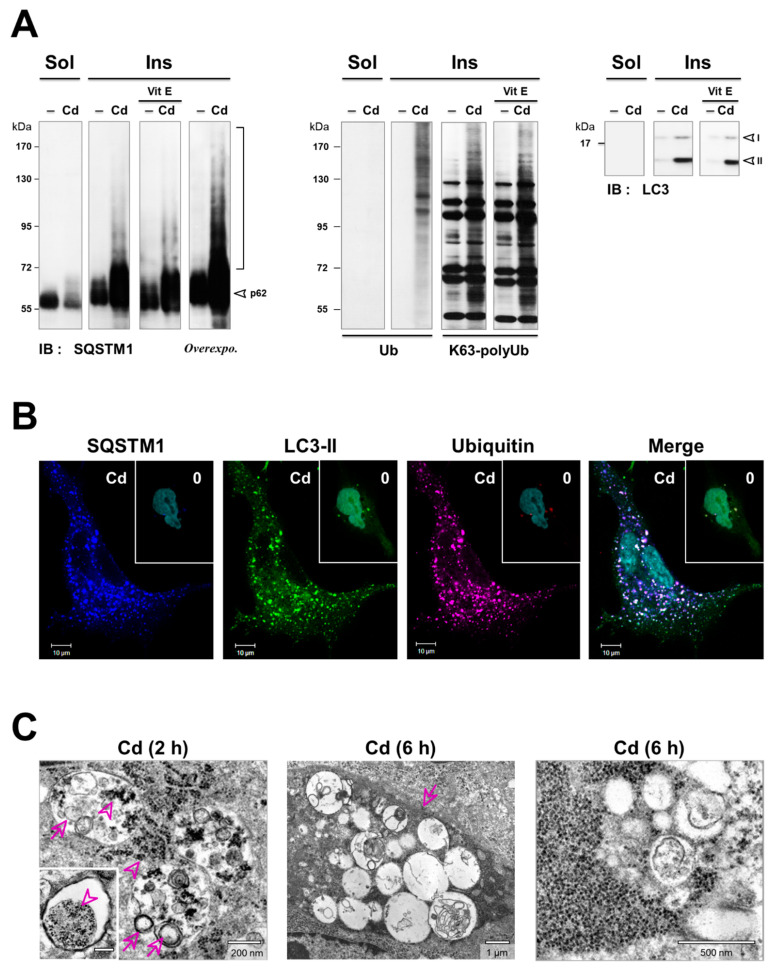
Cd induced accumulation of two selective autophagy markers—SQSTM1 and K63-linked ubiquitinated proteins. (**A**) Accumulation of SQSTM1 and ubiquitinated proteins in the insoluble fractions of Cd-treated cells. A549 lung cancer cell lysates (0.5% Triton-X100) were separated into detergent-soluble (Sol.) and insoluble (Ins.) fractions. Each fraction was examined by immunoblotting with the indicated antibodies. (**B**) Cd induced the formation of ubiquitin-, LC3-II-, and SQSTM1-positive inclusions. Confocal immunofluorescence analysis of Cd-treated A549 lung cancer cells with anti-ubiquitin (magenta), SQSTM1 (blue), and LC3-II (green) antibodies. The right-hand panel shows a merged image. (**C**) Representative electron microscopy images showing the ultrastructure of Cd-treated cells (5 μM, 2 h, and 6 h). Most of the autophagic vesicles range from double-membrane autophagosomes to single-membrane autolysosomes after 2 h. Moreover, all autophagic vesicles contained small electron-dense aggregates, likely ubiquitinated aggregates (arrows). High magnification revealed electron-lucent vesicles clustered at the periphery of inclusion bodies after 6 h. Few, if any, autophagosomes were found in the cytoplasm of untreated cells (*n* = 25). Results are representative of three independent experiments. Twenty to twenty-five EM or IF pictures were taken randomly from each sample. The majority of the cells displayed these modifications. The most representative is presented in the Figure. See Appendix A for uncropped images of immunoblots, Appendix A for electron microscopic photographs of control cells, and Appendix A for electron microscopic photographs of bafA1-treated cells.

**Figure 6 cancers-13-02490-f006:**
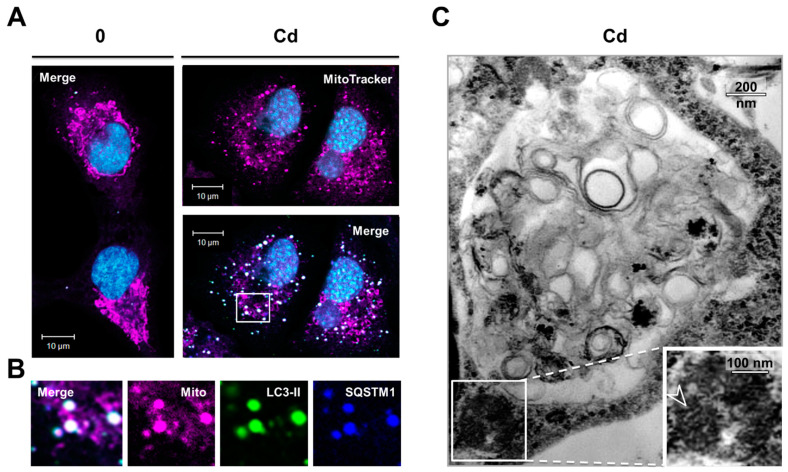
The defect in autophagy induced by Cd was characterized by the formation of inclusion bodies that trap electron-lucent vesicles and mitochondria. (**A**) Shows a representative fluorescence micrograph of control (0) and Cd-treated A549 cells stained with MitoTracker (localizes to mitochondria, magenta), SQSTM1 (detects inclusion bodies, cyan5, blue), and LC3-II (detects autophagic vesicles, green). (**B**) Inset showing clusters of mitochondria colocalized with SQSTM1 and LC3-II in Cd-treated cells in contrast to untreated cells. (**C**) Representative electron microscopy of Cd-treated cells showing the formation of a dense granular aggregate that trapped electron-lucent vesicles and mitochondria (with disorganized cristae, arrowhead) at its periphery. Twenty to twenty-five EM or IF pictures were taken randomly from each sample. The majority of the cells displayed these modifications. The most representative are presented in the Figure. See Appendix A for electron microscopic photographs of control cells.

**Figure 7 cancers-13-02490-f007:**
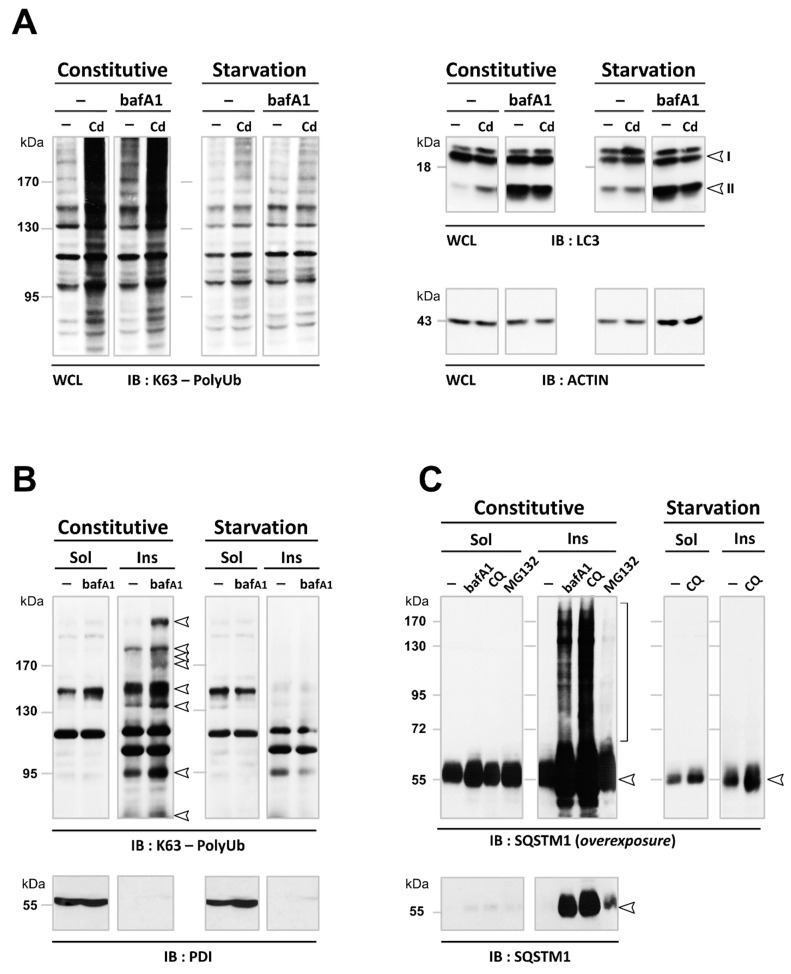
Inhibition of constitutive autophagy initiates the aggregate formation of SQSTM1 and K63-ubiquitinated proteins. Autophagy was inhibited at the degradation step in A549 lung cancer cells using bafilomycin A1 (bafA1, 100 nM) or chloroquine (CQ, 100 µM) in nutrient-rich (constitutive or cons.) or starved conditions (HBSS). (**A**) Accumulation of K63-ubiquitinated proteins in bafA1-treated cells (24 h, whole-cell lysates, WCL). (**B**) Formation of K63-ubiquitinated bodies in bafA1-treated cells. (**C**) SQSTM1 detergent-resistant aggregates in bafA1 and CQ-treated cells. IB of soluble (Sol) and insoluble (Ins) fractions from control bafA1-, CQ-, or Mg132 A549-treated cells (24 h). PDI is a marker for the soluble fraction. Note that K63-ubiquitinated proteins and the Hmw SQSTM1 were visualized only upon inhibition of autophagy (bafA1 or CQ) in nutrient-rich conditions. See Appendix A for uncropped images of immunoblots. (**D**) Degradation of long- and short-lived proteins. PCT cells were metabolically labeled with [^14^C] valine and chased with 10 mM cold valine for either 2 min (labeled short- and long-lived proteins, “Short + Long”) or 16 h (labeled long-lived proteins, “Long”). After labeling and chase, cells were treated with bafA1 (100 nM) in serum- and amino acid-supplemented (constitutive) or -deprived (starvation) media for 5 h. The contribution of autophagy and other lysosomal pathways to the degradation of short- and long-lived proteins is shown. Data are mean ± standard deviation of duplicates and are representative of three independent experiments. Accordingly, similar results were recapitulated in bafA1-treated Sertoli cells, a cell type with a high autophagy flux (Appendix A). * *p* < 0.01, ** *p* < 0.005, and *** *p* < 0.0008.

**Figure 8 cancers-13-02490-f008:**
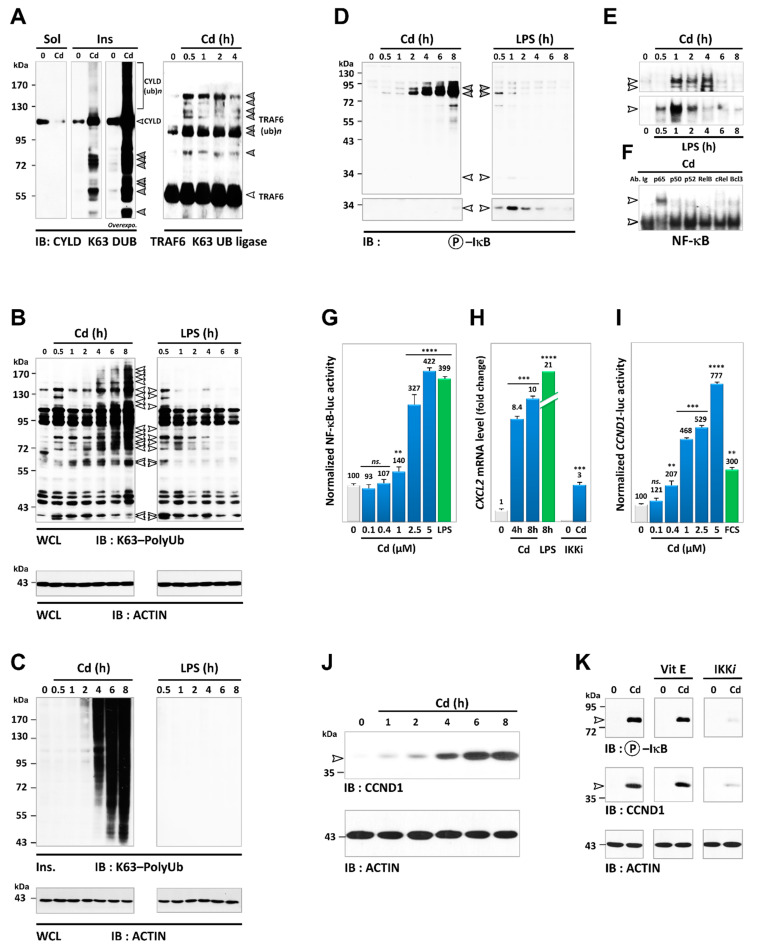
Sustained activation of the NF-κB pathway by Cd. PCT cells were treated with Cd or LPS (5 µg/mL, used as a positive control) for the indicated times in the presence or absence of either vitamin E, the IKK inhibitor (sc-514; 100µM), or vehicle. (**A**) Left: Cd induces the distribution of CYLD from the cytosolic to the insoluble fraction. CYLD immunoblotting shows that CYLD was depleted in the detergent-soluble (Sol) fraction and enriched in the insoluble (Ins) fraction. Note in the insoluble fractions the presence of low molecular weight forms of CYLD, likely cleavage fragments. Right: High molecular weight forms of TRAF6, likely TRAF6 polyubiquitination, were evidenced by TRAF6 immunoprecipitation and immunoblotting of Cd-treated cell lysates. (**B**) Time-course of K63-linked ubiquitination in response to Cd (left) and LPS (right). (**C**) Aggregation of K63-linked ubiquitinated proteins was visible only in the insoluble fractions of Cd-treated cells. (**D**) Accumulation of phospho-IκB (P-IκB) after treatment with Cd or LPS for the indicated time periods. (**E**) A representative EMSA of a time-course of Cd-induced NF-κB activity (Top) and LPS-induced NF-κB activity (Bottom). Two Cd-inducible NF-κB DNA binding complexes were detected. (**F**) Characterization of Cd-induced NF-κB DNA binding complexes with supershift assays using the indicated antibodies. The results are representative of two independent experiments. (**G**) Dose–response effect of Cd on NF-κB luciferase reporter activity (8 h). (**H**) Induction of *CXCL_2_* mRNA upon Cd treatment. (**I**) Dose–response effect of Cd on Cyclin D1 promoter-luciferase reporter activity (8 h). (**J**) Time-course of Cd-induced CCND1 protein expression. (**K**) The IKK inhibitor but not vitamin-E counteracted the activation of the NF-κB pathway by Cd. Results are averages of three experiments ± standard deviation (SD). Not significant (ns), ** *p* < 0.005, *** *p* < 0.0008, and **** *p*< 0.0001. See Appendix A for uncropped images of immunoblots.

## Data Availability

The data presented in this study are available upon request from the corresponding author. The data are not publicly available due to ongoing patent and study.

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
