# Peer review of "The Carcinogen Cadmium Activates Lysine 63 (K63)-Linked Ubiquitin-Dependent Signaling and Inhibits Selective Autophagy"

_cancers, 2021, doi:10.3390/cancers13102490_

Round 1
Reviewer 1 Report
Ref: Cancers-1211623
Title: The carcinogen cadmium activates lysine 63 (K63)‐linked ubiquitin‐dependent signaling and inhibits selective autophagy
Journal: Cancers-MDPI
The manuscript entitled: “The carcinogen cadmium activates lysine 63 (K63)‐linked ubiquitin‐dependent signaling and inhibits selective autophagy” by Chargui et al. is a well written research article that investigates the carcinogen cadmium that subverts selective autophagy, resulting in massive K63‐linked ubiquitination and downstream activation of NF‐kB pathway. The hypothesis and their findings are supported using a variety of experimental methods which further strengthen the manuscript. Moreover, all the comments and suggestions have been addressed by the authors adequately in the revised manuscript. I believe that now the manuscript has been greatly improved and can be considered for acceptance for publication on this journal in the current version. Therefore, I recommend this manuscript to be accepted for publication in the Cancers-MDPI journal.
Reviewer 2 Report
I am satisfied with the majority of the changes and I think the manuscript should now be accepted for publication.
Reviewer 3 Report
I thank the authors for their careful evaluation and experiments following my comments and questions. They perfectly completed manuscript and I do not have any further questions. In my opinion, this manuscript is ready to get published with its present form. Congratulations to them!
This manuscript is a resubmission of an earlier submission. The following is a list of the peer review reports and author responses from that submission.
Round 1
Reviewer 1 Report
Ref: Cancers-1100931
Title: The carcinogen cadmium activates lysine 63 (K63)‐linked ubiquitin‐dependent signaling and inhibits selective autophagy and CYLD K63 deubiquitination
Journal: Cancers-MDPI
The manuscript entitled: “The carcinogen cadmium activates lysine 63 (K63)‐linked ubiquitin‐dependent signaling and inhibits selective autophagy and CYLD K63 deubiquitination” by Chargui et al. is a well written research article that investigates the carcinogen cadmium that subverts selective autophagy and CYLD, resulting in massive K63‐linked ubiquitination and downstream activation of NF‐kB pathway. The authors conclude that their findings link together impairment of selective autophagy, K63‐linked ubiquitination, and carcinogenesis. The present study provides for the first‐time evidence that cadmium (Cd), a widespread environmental carcinogen, is a potent activator of K63‐linked ubiquitination, independently of oxidative damage, activation of ubiquitin ligase, or proteasome impairment. Their findings are supported by different methods and techniques. I recommend this manuscript to be accepted for publication in the Cancers journal, after major revisions. Please find below the few comments-suggestions and major revisions that will help the authors improve the current version of this manuscript:
Major revisions:
-Line 53-54: “Further clarification of these issues will provide valuable insight into prognostic biomarkers and/or therapeutic targets for lung cancer”: you mentioned previously that the study use lung and kidney epithelial cells. Could you please explain why during the further exploration you focus only on lung and not kidney cancer as well?
General comment: The authors need to explain the choice of the focus in lung cancer and not other types and further support their choice of the two epithelial cell lines (lung and kidney) to explore their hypothesis.
Introduction
-Lines 119-120: “degrade signaling proteins, and limit inflammation in normal cells and early stage carcinomas”: please specify these proteins and also explain briefly how inflammation can be linked to cancer progression through these signaling proteins.
Methods
-lines 144-145: “we used mouse PCT (Proximal Convoluted tubules), monkey kidney COS7”: please explain the choice of the mouse PCT and monkey kidney and not a human cell line.
-lines 146-147: “The following Cd doses that produced maximum ubiquitination were chosen…” : please explain how the concentrations within this range (uM) and the specific ones (PCT (Cd 5 μM), COS7 (Cd 10 μM) have been chosen.
-Line 181-190: please be more detailed with the western blot description (i.e how long the antibodies were incubated, how many washes between the incubations etc).
-lines 203-205: “Analyses were performed on a cap‐LC system (Ultimate 3000, Dionex) with pre‐concentration coupled to an Orbitrap Classic mass 204 spectrometer (Thermofisher Scientific).” : please be more specific for the LC/MS and MS/MS conditions (i.e which type of column, running mass spec conditions etc)
-General comment: I would suggest the methodology to be re-structured and presented in terms of techniques (i.e 1. qRT-PCR, 2. Mammalian transfection, 3. Electrophoretic mobility shift assays etc) rathen than topic-related headings (i.e NF‐kB pathway Activation).
-line 208: “To examine the chymotrypsin‐like activity…” : please elaborate why only chymotrypsin activity was examined and not trypsin activity as well. Also, are there other chymotrypsin-like substrates (other than Suc‐LLVY‐Amc) that you could potentially use or have tested?
Results
-line 599: “Cd resulted in sustained activation of the NF‐kB signaling pathway in PCT cells”: it would be good to see whether the activation is linked with changes as well in the gene expression level by analyzing some key gene/proteins of the NF‐kB pathway using qRT-PCR and/or use another experiment for verification.
-“This demonstrated that low doses of Cd activated K63‐linked ubiquitin‐dependent NF‐kB signaling leading to inflammation and cell proliferation”: for the inflammation-related effect it would be better to evaluate more than one inflammatory-related markers (i.e other important cytokines in combination with CXCL2).
-General comment: It would be useful to draw a figure as an illustration that will summarize the full mechanism and the downstream pathways suggested from your study (something similar to your graphical abstract but more enriched with details in the suggested regulatory pathways).
-Figure 7: the figure legend is quite confusing the way it is written; please try to explain the figure with the order that the different parts appear.
-Figure 7(G-I): Please indicate the Y-axis. For instance, maybe you should consider define the levels of expression as fold-changes.
Discussion
-General comment: it would be better in your discussion to include the figures within the text when you describe part of your results.
- “Autophagy switches off the NF‐kB pathway by degrading multiple NF‐kB signaling intermediaries, including IKKα, IKKb, IKKγ, BCL10, and RELA/p65”: it would be interesting to check the expression levels in the current study related to the Cd-induced effect (similar to the above mentioned comment).
-General comment: Results related to the NF-kB pathway should be discussed more in the discussion part (the same for the inflammation-mediated pathways).
-line 692: “exacerbated the production of inflammatory cytokines”: please name a few of those cytokines and briefly explain their role here.
-General comment: in the discussion part please discuss a bit more the link of your suggested model (Cd-induced K63‐linked ubiquitination, independently of oxidative damage, activation of ubiquitin ligase, or proteasome impairment) with different types of cancer progression.
Minor revisions:
-In the graphical abstract the word “carcinogen” at the end should be substituted with “cancer progression” and in general the biological outputs.
-Lines 120-125: please add references to further support the “….as a tumor develops, autophagy drives cancer progression….”
-Lines 128-129: “All the genetic or pharmacological strategies used so far inhibit both catabolic pathways.”: please explain more the term “catabolic pathways” here.
-line 174: “Cells (ø 100)”: please define the cells concentration and write this differently. Also please define the Cd concentrations here as well.
-line 243: “mitochondria as one of the four well-established autophagy substrates”: please replace the word “substrate” since mitochondria is an organelle and cannot be considered as a substrate.
-Figure 1C: Please explain a bit more this part in the figure legend (be more descriptive).
-Figure 1: “The positions of the molecular weight markers are indicated on the left in thousands.”: please replace to “kDa”.
-Figure 2: the mass spec graphs would be better to be aligned in order to be able to compare them and also maybe you could indicate with arrows the important peaks.
-line 379: “but not by other heavy metals (Pb, Hg, Co, Fe)..”: please explain how did you make the choice of these metals and not others (i.e Mg).
-In figure 2 legend it would be better to write them separately as 2B, C, D with different sections since the combination of B-D) looks quite confusing.
-Figure 3D: should be a bit bigger and more visible. I would prefer to see all the figure panels involved in one location (than supplementary). Also, what the Y-axis indicate?
-General comment: A549 lung cell line should be written as : A549 lung cancer cell line.
-line 521: (0.5% Tx‐100): please write the full name of the detergent: Triton X-100 etc.
-Figure 4C: the arrows in the figure would be better to be in color (red) for being more visible.
-line 568: “the activation of NF‐kB by Cd”: please rephrase this as “the Cd-induced activation of NF‐kB pathway”
Reviewer 2 Report
In this study entitled "The carcinogen cadmium activates lysine 63 (K63)‐linked ubiquitin‐dependent signaling and inhibits selective autophagy and CYLD K63 deubiquitination" the authors are assessing the role of exogenous Cadmium addition to the generation of polyK63 chains and the effects it has on autophagy and downstream signalling cascades.
At a first glance I found that the paper was very well written and the methods described in detail. I was very pleased to see the full gels presented in the supplemental figures as opposed to just having a few bands of gels presented. I do feel overall that the figures could be improved in terms of quality.
Fig1B. What is the y axis? Fig 1D. Maybe proteasomal activity (%) should also be on the y axis. Fig 2B,C kDa and markers should be properly labelled on the big blots but also on the actin blots. Fig 3C (and subsequent microscopy containing figures) the authors should move away from red, green, blue and move on to a more colour blind friendly image palette (such as using magenta etc). For Figure 4 and 5 EM staining of untreated cells should be included. Additional other oxidising agents such as H2O2 that the authors found that it does not induce polyubiquitinylation should be included. Also it is not clear to me how many EM images were visualised (or microscopy pictures) and out of how many cells observed the authors could see these modifications. Quantifications should be included to understand if all cells are experiencing these morphological changes or not, at early or later cd time points or high and low cd concentrations. Also better description of figures 4 and 5 should be in the results section as there is no sentence in the text describing 5C for example. Fig 7E better labelling on the figure, also markers everywhere in fig 7 and better annotation is needed.
I would suggest that with these modifications the manuscript will be of better quality.
There is still a lack of understanding on what is cadmium actually doing that perhaps should be discussed further. Additionally, it is not clear to me on what happens first. Maybe a model should be presented at the end of the manuscript as well. In parallel the authors have not looked at lipids. As they are such key components to autophagosome generation etc, I would like to see some lipid staining upon cd addition and lipid oxidation assays. Does cd addition lead to lipid oxidation that can also lead to an impairment of autophagy? The authors should add lipid oxidising hydroperoxides (such as cumene hydroperoxide) as a control in parallel with the cd addition and assess the autophagy phenotypes.
Reviewer 3 Report
Chargui and Belald et al. have shown how cadmium acts on CYLD and selective autophagy, therefore enhances K63-linked ubiquitination and NF-kB pathway during cancer progression models lung and kidney epithelial cells. This study gives important aspects on well-known carcinogenic agent, cadmium, by focusing multiple cellular cascades involving in tumor progression. They provide a lot of important data to support their hypotheses. However, there are still many points to be improved.
Introduction should give more information about the carcinogenic effects of Cd. Instead providing details on well-known cellular processes, they should fuse this knowledge with carcinogenesis of their show cases as they used lung and kidney epithelial cells. And please design your graphical abstract according to the curated results in this manuscript.
In Figure 1, authors should show same time exposures of Cd on oxidized protein levels as Figure 1B. And then they should continue with other experimental setups. That would be nice to see loading control of Figure 1C. What is in vivo statement in Figure 1D? Obviously, they are not using animal models over here. So, they should use the correct terminology. On the other hand, that would be nice to see viability of the cells during Cd exposure. Do they see viability difference? Please show statistical analyses of Figure 1B and Figure 1D. p values should be indicated in the figures or figure legend. However, running citations are making the line of Figure 1 complicated. Please avoid using unnecessary information in the running results.
In Figure 2, if they mention K63-linked ubiquitination enhancing effect of Cd on A459 & PCT cells in the same place. It would be much better for the line of manuscript.
In Figure 3, authors should include p62 into 3A and 3B. Then, it will be much better to evaluate autophagic flux. K63-linked ubiquitination by combination of Cd with HBSS starvation or CQ should be included over here. On the other hand in this figure it would be better to include other markers in terms of selective autophagy types? Please increase the size of Figure 3D.
In Figure 4, quantification is needed for Figure 4B. LC3-II is not correlated with WB data. Normally A459 cells should have autophagic flux under normal conditions (according to literature and the blots in Figure 3). Is Ubiquitin linkage-specific K63? Please define it clearly in the figure. Figure 4C should include controls with E64d+Pep as they were used for Figure 3.
In Figure 5, please use quantifications to show the differences. On the other hand, how does Cd affect mitochondrial depolarization? It should mentioned over here because p62 accumulation. And somehow these figures are missing other cell line which was mentioned at the very beginning as lung and kidney epithelial cells. Where are the experiments of kidney epithelial cells?
In Figure 6, somehow we started to see Sertoli cells, where were the analyses of these cells in the previous figures? Instead of focusing on different cell types in different levels of the manuscript, it is much better to stick on one cell line.
CYLD and A20, these two different DUBs can remove K63-linked ubiquitin chains. CYLD is a really good target but what happens to A20 in Figure 7A? How did they end it up with only CYLD? What happens when they treat cells with TNF-a and Cd? “Cd induced sustained and hyper‐phosphorylation of IkBα that started at 1 h and lasted for 8 h“ if this is the case, what happens NF-kB activation after 4h? And again why did they switch their cells to PCT cells? Did they have same observations in other cells used for previous figures? If selective autophagy is important for NF-kB activation and phosphorylated IkBα, authors should show this mechanistic link in their experimental setup. And the regulation of NF-kB pathway can easily differ between different cell types. So this data should be produced in this context.
Unfortunately this manuscript has many ends. Authors should try to arrange their findings in a more focused way. Otherwise, the message of manuscript is very confusing. Autophagy and NF-kB concepts are already two big fields for this paper. If authors want to combine these concepts , mechanistic links should be clarified. And the bridge experiments should be designed in a better way between figures.
